# Loss of N-terminal acetyltransferase A activity induces thermally unstable ribosomal proteins and increases their turnover in *Saccharomyces cerevisiae*

Ulises H. Guzman[1], Henriette Aksnes [2], Rasmus Ree [2], Nicolai Krogh[3], Magnus E. Jakobsson[1,4], Lars J. Jensen [1], Thomas Arnesen [2,5,6] ✉ & Jesper V. Olsen [1] ✉

Protein N-terminal (Nt) acetylation is one of the most abundant modifications in eukaryotes, covering ~50-80 % of the proteome, depending on species. Cells with defective Nt-acetylation display a wide array of phenotypes such as impaired growth, mating defects and increased stress sensitivity. However, the pleiotropic nature of these effects has hampered our understanding of the functional impact of protein Nt-acetylation. The main enzyme responsible for Nt-acetylation throughout the eukaryotic kingdom is the N-terminal acetyl-transferase NatA. Here we employ a multi-dimensional proteomics approach to analyze *Saccharomyces cerevisiae* lacking NatA activity, which causes global proteome remodeling. Pulsed-SILAC experiments reveals that NatA-deficient strains consistently increase degradation of ribosomal proteins compared to wild type. Explaining this phenomenon, thermal proteome profiling uncovers decreased thermostability of ribosomes in NatA-knockouts. Our data are in agreement with a role for Nt-acetylation in promoting stability for parts of the proteome by enhancing the avidity of protein-protein interactions and folding.

Protein modifications are essential to modulate cellular protein activity, stability, subcellular localization, and interactions[1]. One of the most important protein modifications is acetylation, which can occur co- or post-translationally at the ε-amino group of lysine residues or the free α-amino group at the protein N-terminus[2,3]. The latter, known as Nt-acetylation, is among the most abundant protein modifications in all eukaryotic proteomes[4], but its function remains poorly understood. Advances in high-resolution mass spectrometry and molecular biology techniques have lately helped to shed light on the molecular mechanisms and essential biological processes, where Nt-acetylation

and the enzymes responsible for catalyzing this modification have a central function[5–8].

Chemically, Nt-acetylation refers to a process that involves the covalent addition of an acetyl group to the free amino group of the α-carbon of the N-terminal residue in a protein. This process is catalyzed by Nt-acetyltransferases (NATs) using acetyl coenzyme A (Ac-CoA) as the main donor of the acetyl group[9]. Unlike lysine acetylation, the most exhaustively studied acetylation type, Nt-acetylation has been regarded as an irreversible and static modification, as no deacetylase acting on N-termini has yet been identified[2,4]. Nevertheless, different reports suggested that Nt-acetylation may be regulated by cellular signaling

[1]Novo Nordisk Foundation Center for Protein Research, Proteomics Program, Faculty of Health and Medical Sciences, University of Copenhagen, Copenhagen, Denmark. [2]Department of Biomedicine, University of Bergen, Bergen, Norway. [3]Department of Cellular and Molecular Medicine, University of Copenhagen, Copenhagen, Denmark. [4]Department of Immunotechnology, Lund University, Lund, Sweden. [5]Department of Biosciences, University of Bergen, Bergen, Norway. [6]Department of Surgery, Haukeland University Hospital, Bergen, Norway. ✉e-mail: thomas.arnesen@uib.no; jesper.olsen@cpr.ku.dk

and its cellular status can vary in different disease states and biological processes such as cancer[10], developmental disorders[11,12], drought stress[8,13], calorie restriction and Ac-CoA availability[14,15] or apoptotic fate[16,17]. Thus, Nt-acetylation has emerged as an important protein modification that is involved in the regulation of different biological pathways.

To date, eight NATs have been reported in eukaryotes (NatA-H), of which NatA, NatB and NatC act in a co-translational manner and perform most Nt-acetylation in eukaryotic proteomes[4]. There are well-established examples of how Nt-acetylation may steer protein function: via protein stability and degradation, folding, subcellular localization, and complex formation[18]. The effects of NatA-mediated Nt-acetylation reported so far are very diverse, probably reflecting the large number of NatA substrates[6]. The *Saccharomyces cerevisiae* NatA complex was found to steer gene expression, most likely in part due to Nt-acetylation of the silencers Sir3 and Orc1[7,19,20]. Some impact on protein folding and aggregation was also observed and could result from chaperones directly steered by Nt-acetylation or by the co-operation of chaperones and NatA at the ribosome[7,21,22]. Specific yeast proteins may be targeted for degradation via the exposure of N-degrons. However, global yeast analyses did not reveal Nt-acetylation as a major determinant for protein stability[7,23]. In human and plant cells, a subset of NatA substrates are shielded from proteasomal degradation by Nt-acetylation[17,24,25]. Thus, NatA has the potential to regulate a number of cellular proteins and pathways, but a detailed proteome-wide understanding of how the NatA complex activity may steer proteostasis remains unclear.

For that reason, we applied proteome-wide multidimensional mass spectrometry-based approaches on *Saccharomyces cerevisiae*, lacking NatA complex activity, to explore the link between Nt-acetylation, protein turnover, and thermostability at a proteome scale. Together, the combined analysis of different properties of the proteome suggests that abolishment of NatA complex activity promotes thermal instability of cytosolic ribosomal proteins and increase their turnover. In agreement with previous observations, our results support that Nt-acetylation has an important role in the control of protein stability.

## Results

### Lack of NatA activity induce proteome remodeling in *Saccharomyces cerevisiae*

The NAT machinery in *S. cerevisiae* known to date is composed of five NATs (NatA–NatE). The main contributor to the N-terminal (Nt) acetylome is the NatA complex, which is evolutionarily conserved in eukaryotes[6]. In *S. cerevisiae*, NatA is composed of two essential subunits: a catalytic subunit Naa10 (Ard1), and a ribosome-anchoring auxiliary subunit Naa15 (Nat1) as well as an auxiliary subunit without a well-defined role, Naa50 (Nat5)[6,26,27]. Whereas the substrate specificity of NatB and NatC complexes is determined by the second amino acid after the initial methionine, the NatA complex co-translationally acetylates small amino acids (Ala-, Thr-, Ser-, Val-, and Gly-) at the N-termini exposed after methionine cleavage[18,27–29]. In yeast, NatA is estimated to Nt-acetylate around 40% of the entire proteome[6]. The loss-of-function of Naa10 is embryonic lethal in higher eukaryotes[8,30,31] such as *Arabidopsis thaliana*, *Drosophila melanogaster*, *Danio rerio*, and *Homo sapiens* but not in *S. cerevisiae*. In yeast, deletion of genes encoding either of the major NatA subunits, Naa10 or Naa15, completely abolish NatA activity and cause similar phenotypes[32]. We therefore reasoned that a *naa10Δ* yeast strain would be a suitable model to determine the effect of lacking Nt-acetylation on a proteome-wide scale. Furthermore, the *naa10Δ* strain allows the study of the Nt-acetylome and proteome concurrently, circumventing the compensatory effects of the Nt-acetylation backup systems described in mice and in human[30,33].

The *naa10Δ* strain used here replicated the previously described phenotypic responses to stressors and we found that loss of Naa10 had a negative impact on cell growth in synthetic complete liquid medium (Supplementary Fig. 1A). To be able to compare WT and *naa10Δ* strains at similar growth stages, we compensated for the slower cell doubling time by harvesting WT and *naa10Δ* cells when they reached an optical density at 600 nm of ~1.8. To determine the effect of the loss of Naa10 on the yeast proteome, we performed a differential global protein expression analysis comparing the proteome differences between WT and *naa10Δ* strains. As described in "Methods" and schematized in Fig. 1A, we made use of offline high-pH reversed-phase chromatography to fractionate peptide mixtures resulting from tryptic digestion of yeast lysates prior to online low pH LC–MS/MS analysis of each fraction in turn. The comprehensive yeast proteome presented in this study contains 4113 and 3943 protein-coding genes for WT and *naa10Δ* strains, respectively (False discovery rate; FDR < 1%), which represents the detection of about 96% of the proteome expected to be expressed during log phase[34–36] (Fig. 1B). With this proteome depth, we were able to detect all four NATs (NatA, NatB, NatC, and NatE) expressed during log-phase growth in *S. cerevisiae* (Supplementary Fig. 1B). Unsurprisingly, we did not detect Naa40 (NatD), as it has been reported not to be expressed during log-phase growth[35]. As expected, the catalytic subunit of the NatA complex Naa10 was only present in the WT and not detected in the *naa10Δ* strain. Noteworthy, the other components of the NatA complex, Naa15 and Naa50 (NatE), were also significantly down-regulated in the KO condition. This indicates that lack of Naa10 disrupts the NatA complex formation resulting in degradation of the Naa15 and Naa50 subunits, in agreement with previous data[26]. Conversely, the NatB complex subunits Naa25 (Mdm20) and Naa20 (Nat3) as well as the NatC complex subunit Naa30 (Mak3) remained unaltered in the *naa10Δ* strain (Supplementary Fig. 1B).

The differential proteome expression analysis between WT and *naa10Δ* strains highlighted that some members of the Arg/N-end rule and ubiquitin-fusion degradation (UFD) pathways UBR1, UFD4, UFD2, NTA1, and TOM1 together with the proteasome and autophagy markers such as ATG1, ATG20, ATG11, and PEP4 were upregulated in the KO. In contrast, cytosolic and mitochondrial ribosomal proteins from the small and large subunit as well as mitochondrial proteins related to the electron transport chain were down-regulated (Fig. 1C and Supplementary Fig. 1C). In addition, the classification of the regulated proteome by gene ontology (GO) analysis, showed a significant downregulation of proteins associated with the cellular component terms mitochondrion, cytosolic and mitochondrial ribosomes in the *naa10Δ* strain (Fig. 1D). To investigate this observation further, we performed a gene set enrichment analysis (GSEA) using KEGG pathways (Supplementary Fig. 1D). The GSEA analysis revealed an overrepresentation of autophagy and endocytosis pathways in the *naa10Δ* strain, while KEGG pathways related to metabolic regulation, ribosomal structural proteins, and oxidative phosphorylation were underrepresented. These findings are in agreement with previous reports linking the loss of NatA activity with impairment of mitochondrial degradation[37], a general effect on genome stability and metabolism, as well as the upregulation of the UPS system upon loss of Nt-acetylation[7,38].

### Effect of abolished NatA-mediated Nt-acetylation on protein half-lives

The downregulation of ribosomal proteins observed in the *naa10Δ* strain suggests that Nt-acetylation negatively affects the stability of these proteins. Earlier studies in *S. cerevisiae* point to Nt-acetylation as a mechanism for steering protein degradation as part of cellular quality control[39–41], while other investigations did not uncover any major impact on protein stability or degradation[7,23]. Thus, to elucidate the role of Nt-acetylation on protein half-lives, we reasoned that a systematic and proteome-wide investigation was needed. Global

protein half-life were estimated using a pulsed stable isotope labeling by amino acids in cell culture (pSILAC) chase labeling approach quantifying the incorporation of light ($^{12}C$,$^{14}N$-enriched) stable isotope labeled lysine, an essential amino acid, into newly synthesized proteins, while pre-existing proteins remain in the pre-labeled heavy stable isotope ($^{13}C$,$^{15}N$) form[42]. We analyzed populations of WT and *naa10Δ* strains growing in log phase and sampled them at six different time points, corresponding to approximately two- and three-cell doubling times (Fig. 2A). Protein extracts were digested with endo-proteinase Lys-C to ensure quantification of resulting peptides, which were independently fractionated into 12 fractions by offline high-pH reversed-phase chromatography and each fraction was measured by online LC–MS/MS. A combined analysis identified 47,270 peptide sequences and 4333 proteins (FDR < 1%) across all experimental conditions. As protein degradation generally follows first-order kinetics, assuming that a newly synthesized protein has the same probability of being degraded as a pre-existing, protein loss follows an exponential decay, and the log-transformed relative isotope abundance (RIA) is therefore expected to display a linear behavior with a negative slope in

the time domain[43]. Based on this, we were able to determine 3420 protein half-lives ($T_{1/2}$) with high confidence inferred from the calculated slopes of linear regression (Supplementary Data 1). In addition, we were able to determine the half-life for 570 Nt-peptides, 411 for WT and 365 for *naa10Δ* (Fig. 2B and Supplementary Data 2).

Since the balance between protein degradation and synthesis is a regulated process involved in the coordination of multiple cellular responses, including cell signaling, cell cycle progression, etc.[44,45], protein synthesis and degradation rates depend on cellular surveillance systems. Two main processes determine protein half-life: (i) dilution due to cell growth and (ii) intracellular degradation via the proteasome or lysosome[46,47]. While degradation is a selective process regulating protein half-life in a specific manner[48], the dilution is a global process effectively reducing the cellular protein amount by fifty percent for every cell cycle[49]. Thus, to account for the effect of the dilution on protein half-life, the growth rate of WT and *naa10Δ* strains were determined by estimating the median of RIA at each timepoint. The reason for determining the cell cycle rate using this strategy instead of optical density was that the difference in cell size between

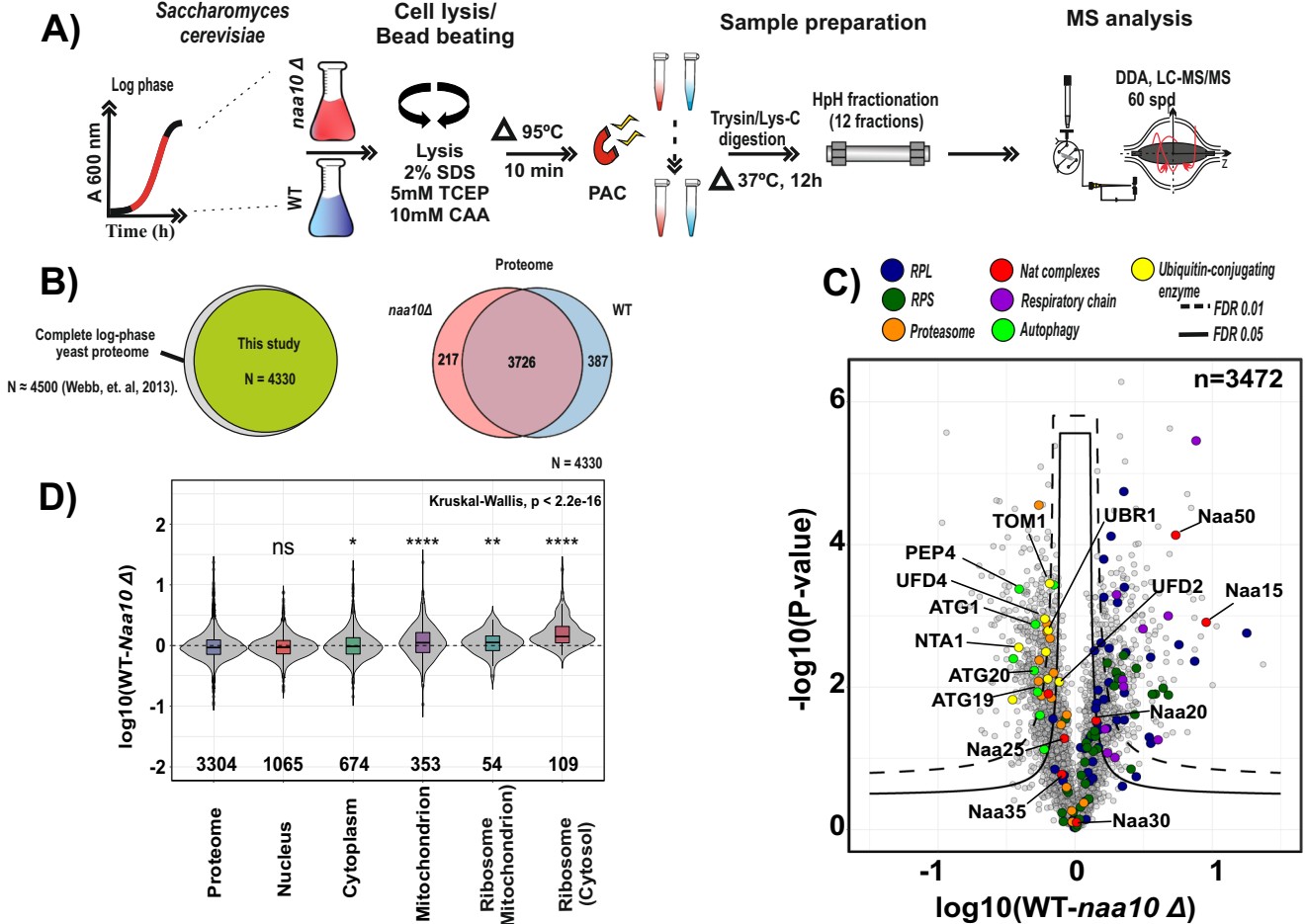

**Fig. 1 | Lack of NatA activity (naa10Δ) causes the downregulation of mitochondrial and ribosomal proteins. A** Scheme of the performed proteome profiling experiments. Yeast were harvested at active growth phase and lysates were digested using the proteome capture aggregation method (PAC) prior High-pH fractionation (HpH). Quantification was performed by using label-free intensities (LFQ; label-free quantitation). *n* = 3 replicate cultures per condition. **B** Right: Venn diagram depicting all proteins quantified in the deep proteome profiling from this dataset compared to complete yeast log-phase proteome reported in ref. 36. Left: Venn diagram depicting numbers of all proteins quantified in the deep proteome profiling, (Proteome, *n* = 4330). **C** Differential expression profiling of the WT and *naa10Δ* strains in a volcano plot. Significant regulated proteins at 1% and 5% false

discovery rate (FDR) are delimited by dashed and solid lines, respectively (FDR controlled, two-sided *t* test, randomizations = 250, s0 = 0.1). Ribosomal proteins from large (RPL) and small subunit (RPS) are colored in blue and green, respectively. **D** Violin plot of the cellular component (CC) gene ontology annotation of the log10 fold detected proteome ratios. Kruskal–Wallis test with two-sided Wilcoxon post hoc test, multiple test correction according to Benjamin–Hochberg, ns *P* > 0.05, *P* <= 0.05, **P* <= 0.01, ***P* <= 0.001, ****P* <= 0.0001; box bounds correspond to quartiles of the distribution (center: median; limits: 1st and 3rd quartile; whiskers: +/− 1.5 IQR). (*n* = 3, *n* = independent cultures per condition). Source data are provided as a Source Data file (**B**–**D**).

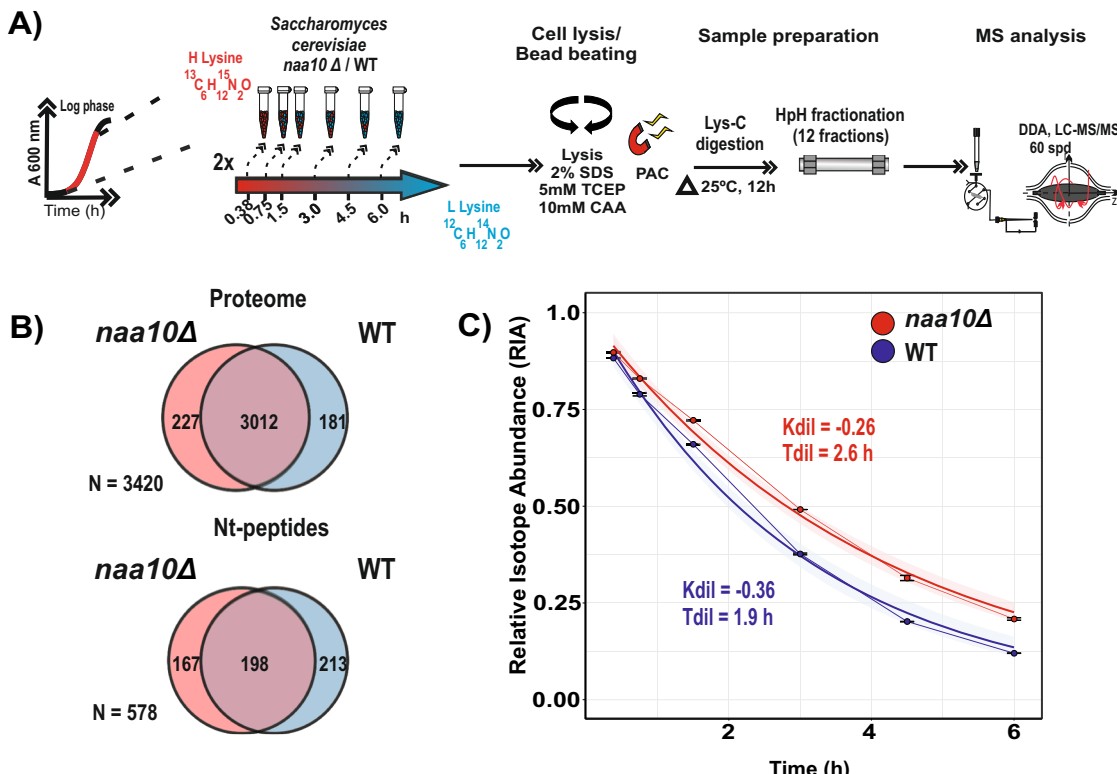

**Fig. 2 | Protein half-life and cell cycle time inference by a pulsed-SILAC chase (pSILAC). A** Schematic representation of the implemented Pulsed-SILAC chase strategy (pSILAC). Yeast cells grown in heavy lysine were pulse-labeled with light lysine and lysed after six time points. Lysates were digested using only Lys-C as protease prior to High-pH fractionation (HpH). H/L SILAC ratios were quantified, and the Relative Isotope Abundance (RIA) calculated. $n = 2$ replicate cultures per condition. **B** Venn diagram condensing numbers of all protein and Nt-peptides inferred the half-lives inferred by pSILAC (Proteome, $n = 3420$, N-terminome, $n = 578$). **C** Time-domain RIA incorporation into *naa10Δ* and WT system proteins. Mean ± standard deviations are shown. The WT and *naa10Δ* calculated dilution constant (Kdil) and dilution time (Tdil; cell double time) are shown in blue and red, respectively. ($n = 2$, $n =$ independent cultures per condition). Source data are provided as a Source Data file (**B**, **C**).

the *naa10Δ* and WT would introduce errors in the OD measurements and also because this technique does not distinguish between living and potentially dead cells. By calculating the dilution constant ($K_{dil}$) for WT and *naa10Δ* strains, we determined that the growth rate of the *naa10Δ* strain is 28% slower than the corresponding WT strain (Fig. 2C).

To establish if the observed growth rate difference between *naa10Δ* and WT strains strongly affected the half-life estimations, we modeled the effect of the 28% decrease of $K_{dil}$ in the *naa10Δ* on the WT protein half-lives (Supplementary Fig. 2). The decrease in growth rate shows that proteins with a longer half-life became even longer-lived, whereas short-lived proteins were unaffected (Fig. 3A). The fact that generally long-lived proteins became even longer-lived suggests that delay in the cell cycle doubling time acts as confounder in comparison of protein half-life estimations between conditions as an apparent stabilization of proteins of long-lived proteins are determined mainly by the difference in growth rate[46,50,51]. Thus, to explore the effect of NatA-dependent Nt-acetylation on protein half-life, we normalized the turnover rate by the corresponding growth rate for WT and *naa10Δ* strains, respectively (Fig. 3B). Using this strategy, we observed that the normalized turnover rates of proteins are generally faster in *naa10Δ* compared to the WT. Next, we compared the total distribution of the normalized turnover rates of proteins with the corresponding Nt-peptides determined in WT (Fig. 3C) and *naa10Δ* strains (Fig. 3D). Interestingly, we found no statistical difference between the normalized turnover rates of the proteome compared to Nt-peptides in WT, but there was a statistically significant difference in the corresponding turnover rates in *naa10Δ* cells. This observation is to be expected since

the inferred half-life of Nt-peptides from pSILAC experiments corresponded mostly to acetylated substrates in the WT and non-acetylated NatA in *naa10Δ* cells, respectively. These findings were further confirmed by comparing the normalized turnover rate distribution of NatA substrates from the *naa10Δ* strain against NatA substrates from WT strain and *naa10Δ* full proteome (Supplementary Fig. 3A, B). Finally, to investigate if the lack of NatA and thus Nt-acetylation of NatA substrates affect their turnover rates in the *naa10Δ* strain, we compared Nt-acetylated peptides identified in WT against the corresponding free Nt-peptides in the *naa10Δ* strain, representing NatA type substrates, which start with Ala Thr, Ser, Val, or Gly at position 2+. This analysis revealed that the free Nt-peptides have a significantly faster turnover rate compared to the Nt-acetylated peptides (Fig. 3E, Supplementary Fig. 3C, and Supplementary Data 3). This finding indicates that N-terminal protein acetylation is a modification that promotes protein stability rather than instability in the yeast proteome.

In addition, to corroborate the Nt-acetylation status of proteins in the *naa10Δ* strain, we took advantage of the high protein sequence coverage provided by the high-pH fractionation in the pSILAC experiments. We specifically analyzed the last timepoint of essentially full light SILAC incorporation and complemented the detected Nt-peptides with Nt-peptides detected by a specific N-terminal enrichment strategy (Supplementary Fig. 4A). The reason for this was that the pSILAC experiment and the N-terminal enrichment were digested with complementary enzymes (Lys-C and Arg-C like digestion with trypsin, respectively) allowing increased coverage of the N-terminome. Collectively, we identified 1480 non-redundant Nt-peptides (Supplementary Fig. 4B) considering only those covering the first or second amino

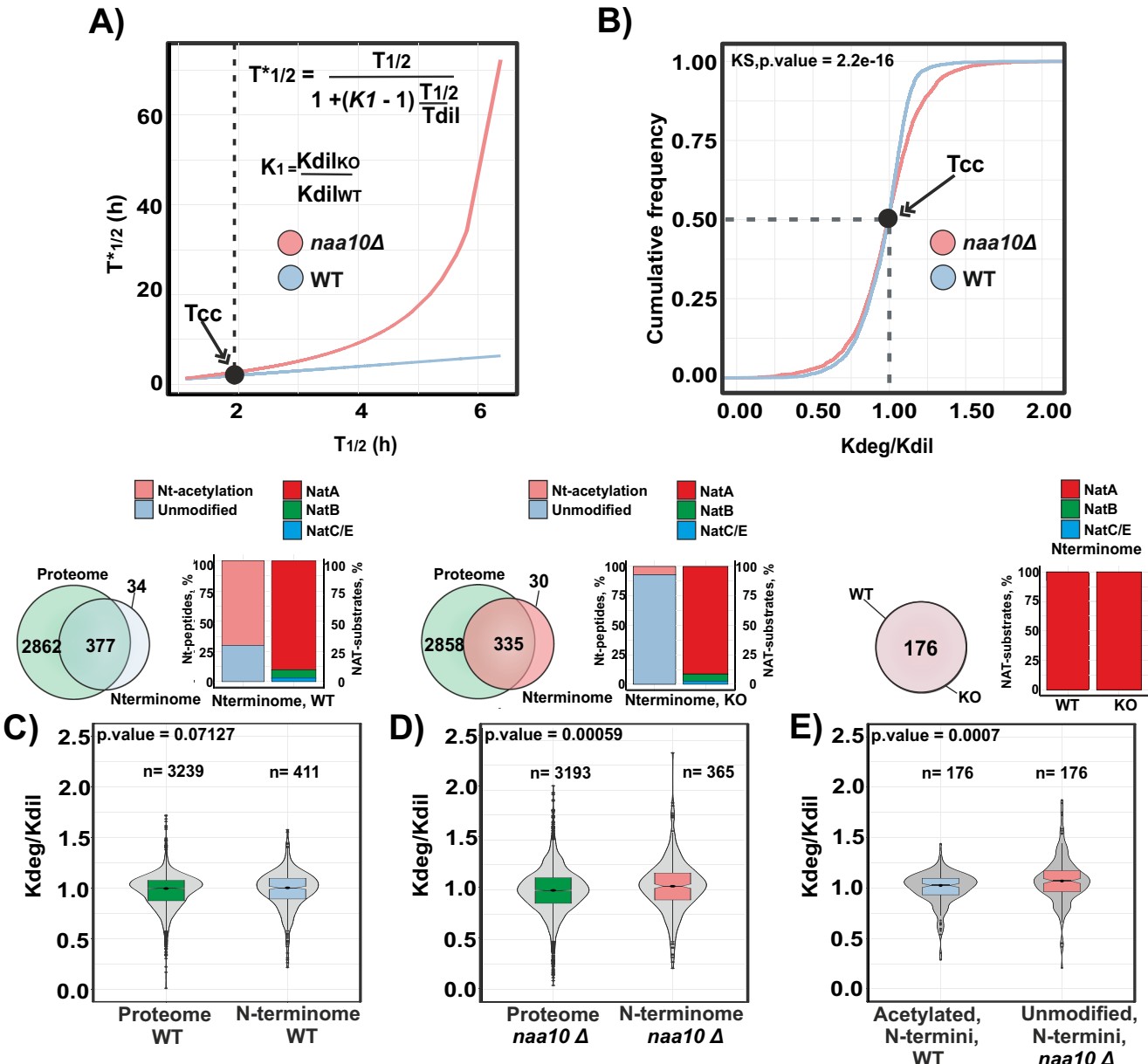

**Fig. 3 | Lack of N-terminal acetylation due to deletion of NAA10 promotes protein degradation of NatA substrates in the yeast proteome. A** A comparison between determined half-lives in WT system ($T_{1/2}$) and modeled effect of WT half-lives ($T^*_{1/2}$) with a reduced dilution constant ($Kdil_{KO} = -0.26$). Red line represents the predicted half-lives due to the lack of *naa10* and blue line the determined half-lives in WT. Cell cycle time ($Tcc = 1.9$ h) of the WT system is marked by an arrow (Black dot). Data were modeled using the equation show on top and calculated according to ref. 46. **B** Cumulative frequency plot of the normalized turnover rate (Kdeg/Kdil) determined in *naa10Δ* and WT system (WT, blue; *naa10Δ*, red). Two-sided Kolgomorov-Smirnov test, KS $P = -2.2$ e-16. **C, D** Violin plot of normalized turnover rates of WT and *naa10Δ* proteome compared to their corresponding

N-terminome. Statistical significance was assessed using two-sided Wilcoxon test, multiple test correction according to Benjamini–Hochberg, ns = $P > 0.05$, $*P <= 0.05$, $**P <= 0.01$, $***P <= 0.001$, $****P <= 0.0001$; box bounds correspond to quartiles of the distribution (center: median; limits: 1st and 3rd quartile; whiskers: +/− 1.5 IQR). Overlap between the N-terminome and proteome detected in the pSILAC, as well as the N-terminome acetylation status and NAT substrate class are shown on top. ($n$ = protein or peptide normalized turnover rates derived from at least two independent experiments per condition). **E** Same as (**C, D**), but comparing the N-terminal acetylated peptides of the NatA type detected in the WT and their corresponding unmodified N-terminal peptides detected in *naa10Δ* cells. Source data are provided as a Source Data file (**A–E**).

acid of annotated protein sequences retrieved from the UniProt database[52] depending on the cleavage of the initiating methionine (Met[i]). We found that 73% of the WT N-terminome was acetylated and 60% of those acetylated peptides were substrates of the NatA complex (Supplementary Fig. 4C), in agreement with previous studies[6,15,29]. Moreover, the *naa10Δ* strain showed a reduction of 56% of the acetylated N-terminome, which fits well with the proportion of the N-terminome acetylated by the NatA complex (Supplementary Fig. 4D). Reassuringly, most of the unmodified Nt-peptides detected in

the *naa10Δ* strain found to be Nt-acetylated in WT, matched the known NatA substrate motif of N-terminal Ala-, Thr-, Ser-, Val-, and Gly- after the Met[i] has been removed by methionine aminopeptidases. In contrast, Nt-peptides representing NatB type substrates, thus starting with Met[i] before Asp-, Asn-, Glu- or Gln- at position +2 were unaffected between samples (Supplementary Fig. 4E). Furthermore, our results correlate with the reported likelihood of a protein to being Nt-acetylated based on the +2 amino acid in its sequence, where proteins with Ser- and Thr- at +2 have a higher probability to be acetylated

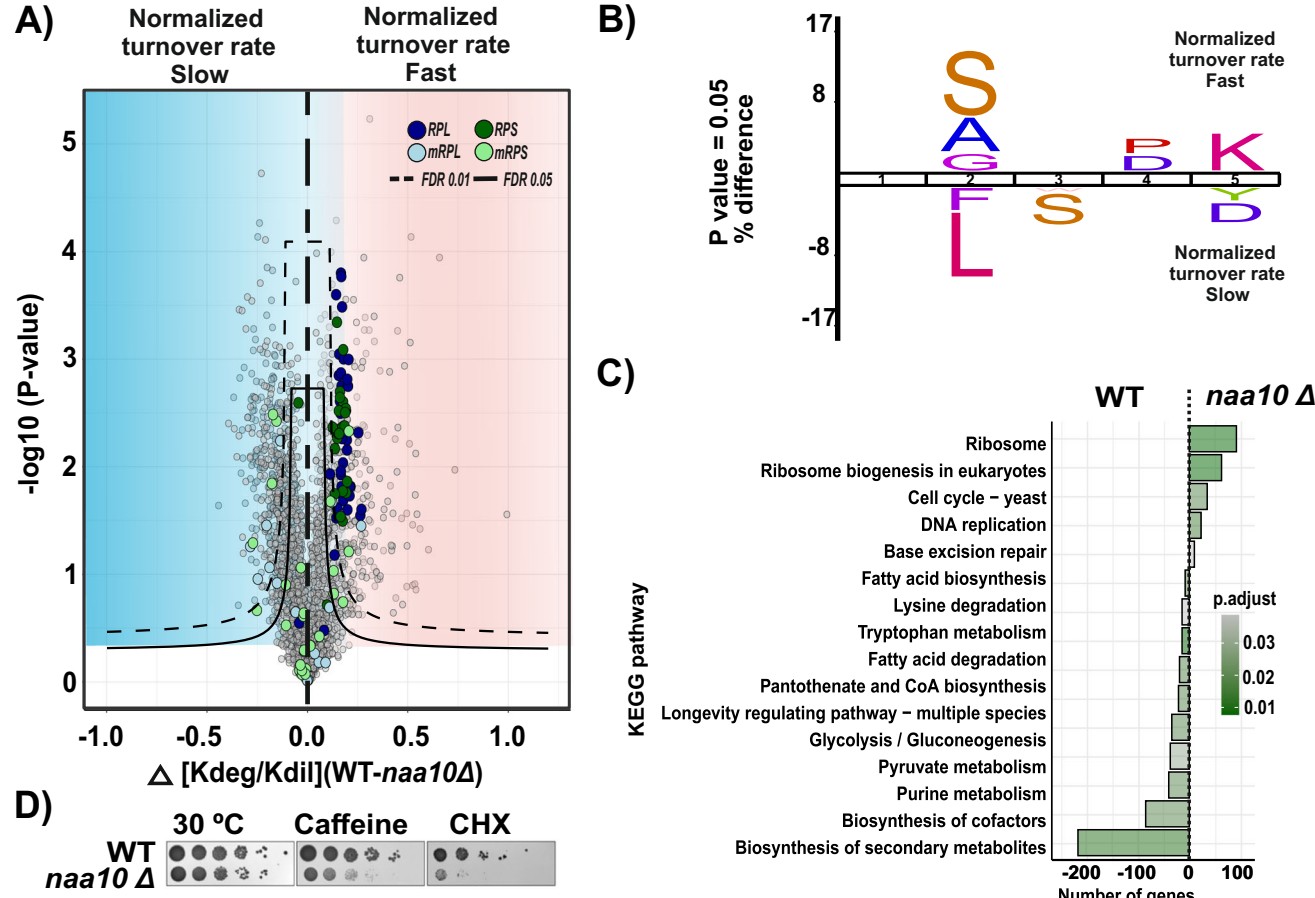

**Fig. 4 | Lack of NatA activity promotes the enhanced degradation of cytosolic ribosomal proteins. A** Comparison of the *naa10Δ* and WT systems as a volcano plot to identify significant changes in protein turnover rates. Significant regulated proteins at 1% and 5% false discovery rate (FDR) are delimited by dashed and solid lines, respectively (FDR controlled, two-sided *t* test, randomizations = 250, s0 = 0.1). Significant proteins were classified into two groups; Fast, red and slow, blue. **B** Ice logo diagram comparing the first five sequence amino acids of the significant fast and slow normalized turnover rate proteins (*P* < 0.05 unpaired two-tailed Student's *t* test). **C** GSEA-based KEGG pathway enriched analysis. *P* values were calculated by a two-sided permutation test and multiple hypothesis testing was FDR corrected. The significance threshold set at FDR > 0.05. **D** Phenotypic growth of *S. cerevisiae naa10Δ* in the presence of various stressors. The indicated yeast strains were grown to early log phase and serial 1/10 dilutions containing the same number of cells were spotted on various media and imaged the 6 following days. 30 °C, incubated for 3 days on YPD at 30 °C; Caffeine, incubated for 3 days on YPD + 0.2% caffeine; and CHX, incubated for 3 days on YPD + 0.2 μg/ml cycloheximide. Source data are provided as a Source Data file (**A**).

compared to the ones with Gly- and Val- at 2+[29] (Supplementary Fig. 4F and Supplementary Data 4).

## Absence of NatA-dependent N-terminal acetylation increases the turnover rate of ribosomal proteins in exponentially growing yeast cells

To facilitate the analysis of changes on turnover rate between the experimental conditions, we decided to visualize them using a volcano plot analysis and classified the *t* test significant proteins into two groups designating if they had faster or slower turnover rate in the *naa10Δ* strain compared to the WT (Fig. 4A). Interestingly, cytosolic ribosomal proteins have faster turnover rates compared to the WT, while the mitochondrial ribosomal proteins show mixed behavior.

In addition, we analyzed the sequence motif consensus of the first five amino acid residues from the proteins classified as significant in the two groups (Fig. 4B). Potential NatA substrates with the amino acid residues Ser-, Ala-, and Gly- in the second position were significantly overrepresented in the fast turnover rate group, suggesting that the absence of Nt-acetylation due to loss of NatA activity in the *naa10Δ* strain is responsible for the fast degradation of these proteins. This result supports that Nt-acetylation promotes protein stability and underpins the importance of this modification across eukaryotes.

Conversely, potential NatC (or NatE) substrate proteins with Phe and Leu in the second position (Met-Phe and Met-Leu N-termini) seem to have a slower turnover rate in the *naa10Δ* strain compared to WT, indicating that the turnover of these proteins are not directly affected by the lack of NatA but via downstream effects. The over-representation of substrates of different NAT types, according to the turnover classification, could suggest that the specificity of the different NATs is connected to different biological functions, as also indicated previously[7]. We confirmed this observation by subgrouping the *naa10Δ* strain proteome into NatA-C substrates and compared the normalized turnover rates of each subgroup against the *naa10Δ* full proteome normalized turnover rates (Supplementary Fig. 3D–F).

To further substantiate this observation, we performed a GSEA-based KEGG pathway enrichment analysis of the fast and slow turnover rate protein groups (Fig. 4C). We found that the faster-degraded proteins are enriched for ribosome and ribosome biogenesis pathways. Contrastingly, the slower degrading proteins are enriched for members of pathways related to the biosynthesis of secondary metabolites, purine metabolism, lysine degradation, and acetyl-CoA and fatty acid metabolism.

Noteworthy, several cytosolic ribosomal proteins belonging to the 60 S and 40 S subunits have been annotated as substrates of the

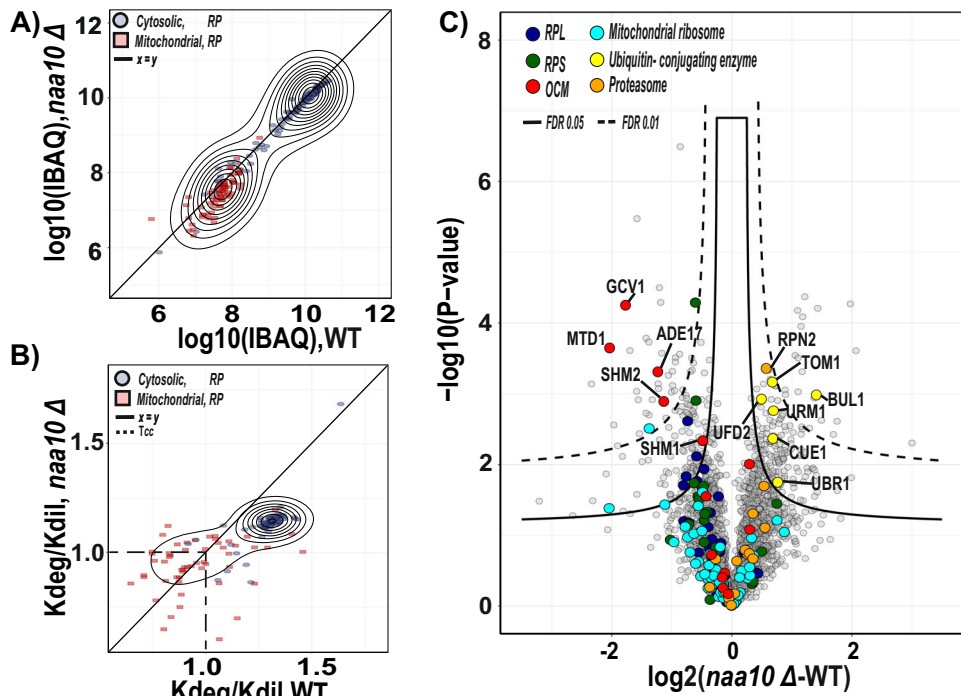

**Fig. 5 | NatA-defective yeast cells compensate for fast ribosomal turnover rate by adjusting protein synthesis. A** Scatterplot comparing log10 of the intensity-based absolute quantification (IBAQ) of cytosolic and mitochondrial ribosomal proteins between WT and *naa10Δ* strains. *n* = 3 replicate yeast cultures per condition. **B** Same as (**A**) but comparing the normalized turnover rate between conditions. *n* = 2 replicate cultures. Diagonal line in (**A**, **B**) correspond to the identity function. Dashed lines indicate the cell cycle time (Tcc). **C** Comparison of the WT and *naa10Δ* strains as a volcano plot to identify newly synthesized protein abundance significant changes. Significant regulated proteins at 1% and 5% false discovery rate (FDR) are delimited by dashed and solid lines, respectively (FDR controlled, two-sided *t* test, randomizations = 250, s0 = 0.1). Source data are provided as a Source Data file (**A**–**C**).

NatA complex[53] (Supplementary Data 5). However, the functional effects of the absence of Nt-acetylation in the ribosome are not well understood. In general, Nt-acetylation has been associated with decreased protein synthesis of the ribosome components and its assembly. Interestingly, in the specific case of NatA-deficient cells, polysome profiling experiments have shown a normal 60 S/80 S ratio[7,53], but a decrease in translational fidelity in the presence of protein synthesis inhibitors in *naa10Δ* strain compared to WT[53]. Thus, this suggests that the reduced translational fidelity in NatA-deficient cells is most likely due to defective activity or structure of fully assembled 80 S ribosome. In support of this, the *naa10Δ* strain exhibits normal ribosome biogenesis, addressed by northern blotting analysis of pre-rRNA processing intermediate (Supplementary Fig. 5A–C). Likewise, the *naa10Δ* strain showed increased sensitivity to the protein translation elongation inhibitor cycloheximide and caffeine (Fig. 4D)[6]. However, these effects of Nt-acetylation on ribosomes are challenging to define, since the fast degradation of ribosomal proteins in the *naa10Δ* strain could conceivably be an indirect consequence of the fast degradation of NatA substrates or other dysfunctionality caused by lack of Nt-acetylation of ribosomal regulatory proteins.

**Newly synthesized ribosomal proteins maintain their protein levels at log phase despite their increased turnover rate**

In *S. cerevisiae*, it has been reported that protein synthesis rate decreases with decreasing growth rates[54] and exponential growth rates require ribosome synthesis[55]. Consequently, for balancing the growth rate at log phase, we speculated that the slower growing *naa10Δ* strain needs to adjust the synthesis of ribosomal proteins to compensate for their faster degradation due to the lack of Naa10. To investigate this, we estimated the relative abundance of the ribosomal and ribosome-associated proteins between WT and *naa10Δ* by intensity-based

absolute quantification (iBAQ) analyzing only the light stable isotope labeled lysine-containing peptides at 6 h after the SILAC pulse as a proxy of the abundance of newly synthesized proteins (Fig. 5A). This revealed that the relative abundance of the newly synthesized cytosolic and mitochondrial ribosomal proteins did not change between the WT and *naa10Δ* strains. This observation is in agreement with previous reports showing that protein levels in NatA-depleted eukaryotic models tend not to differ compared to WT at the mid-log phase. However, the translation rate increases under physiological conditions[7,24], pointing to the fact that protein synthesis rates are adjusted in NatA-deficient cells to maintain a functional concentration of ribosomes.

In contrast, when comparing the turnover rate of ribosomal proteins, those annotated as belonging to the cytosolic ribosomes have a faster turnover rate in the *naa10Δ* compared to WT cells (Fig. 5B). Conversely, the turnover rates of mitochondrial ribosomal proteins showed a mixed behavior suggesting that enhanced synthesis of mitochondrial ribosomal proteins is induced as a response of impaired mitochondrial function. This phenomenon has also been described in previous ribosome profiling and RNA-seq studies, which showed elevated translation of nuclear-encoded genes of mitochondrial ribosomal proteins in NatA-lacking strains[7,53,56].

To visualize the global proteome abundance change of newly synthetized proteins between *naa10Δ* and WT strains, we performed a volcano plot analysis (Fig. 5C). This revealed that proteins related to the ubiquitin–proteasome system (UPS), such as TOM1 and UFD2 were upregulated in the *naa10Δ* strain, which aligns well with the steady-state proteome analysis (Fig. 1B). UFD2 is a member of the ubiquitin-fusion degradation (UFD) pathway, which elongates ubiquitin moieties by lysine-ε-amino specific linkage of ubiquitin N-terminal leading to their degradation by the proteasome[39]. Likewise, the TOM1 protein, which is the homolog of human HUWE1, has

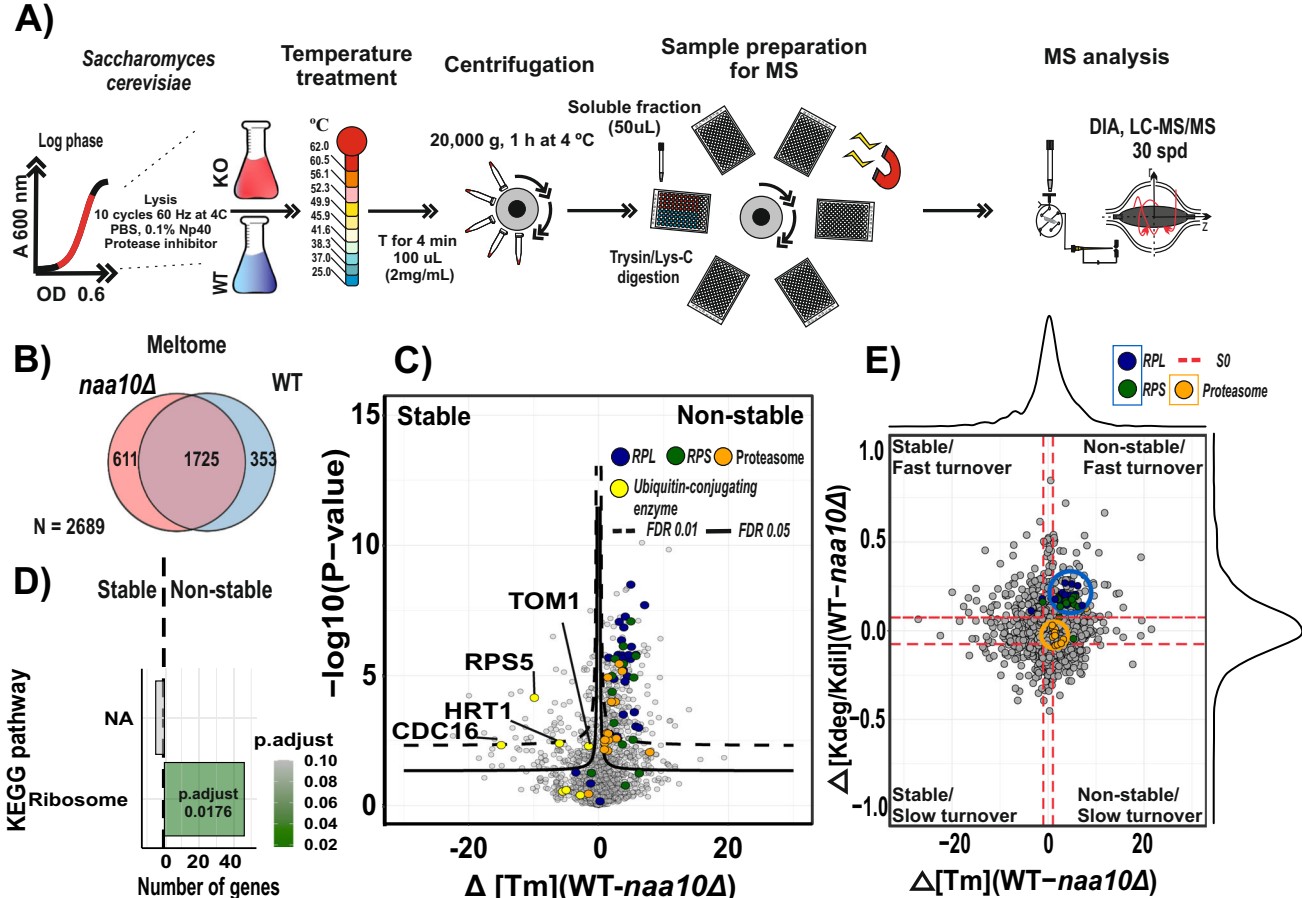

**Fig. 6 | Ribosomal proteins are unstable and rapidly degraded in NatA-defective yeast cells.** **A** Schematic representation of the implemented TPP-DIA strategy. Yeast cells grown in mid-log phase were harvested by centrifugation and submitted to a temperature treatment. Lysates were digested using Lys-C and trypsin. Quantification was performed by using label-free intensities (LFQ; label-free quantitation). $n = 6$ replicate cultures per condition. Melting curves were inferred by using a four-parameter log-logistic model. **B** Venn diagram depicting numbers of all protein-melting temperatures, (Meltome, $n = 2689$). **C** Comparison of the WT and *naa10Δ* strains in a volcano plot to identify changes in protein-melting temperature. Significant regulated proteins at 1% and 5% false discovery rate (FDR) are delimited by dashed and solid lines, respectively (FDR controlled, two-slide $t$ test, randomizations = 250, s0 = 0.1). **D** GSEA-based KEGG pathway enriched analysis. $P$ values were calculated by two-sided permutation test and multiple hypothesis testing was FDR corrected. Significance threshold set at FDR > 0.05 Significance threshold set at $P$.adjust <0.05, NA: no enriched terms at the specified cutoff. **E** Scatterplot representing the overlap between protein thermostability and degradation. Dashed red line delineate the minimal fold change (s0, 0.1). Melting temperature and degradation difference distribution are shown in the top and right plot border, respectively. Source data are provided as a Source Data file (**C**, **E**).

been linked to the degradation of excess histones[57,58], ribosomal proteins made in excess and thus not assembled into mature ribosomes[59] and different pre-replicative complexes during $G_1$[60]. Recently, it was shown that TOM1 co-immunoprecipitates with UFD-like substrate reporters, linking TOM1 with UFD pathway and degradation in *NAA10*-deficient cells[61]. In addition, TOM1 has been linked to a novel quality control pathway, which governs the homeostasis of ribosomal proteins. This pathway is important for responding to imbalances in production of ribosome components, which, if it is not regulated, can exacerbate the temperature-sensitive growth[53] and precipitation of ribosomes[59], a phenotype already described for the *naa10Δ* strain[7]. The differential analysis of newly synthesized proteins also disclosed that proteins related to one-carbon metabolism (OCM) were down-regulated in the *naa10Δ* strain (Fig. 5C). Interestingly, OCM protein members have been implicated in the regulation of the crucial steps of protein synthesis, growth, and translation processes[62,63]. This observation is in agreement with the *naa10Δ* phenotype as the OCM has been linked with the control of protein synthesis through the regulation of the abundance of key substrates such as formylated methionine (Met-tRNA$^{fMet}$) and amino acids[64].

## Absence of NatA-dependent N-terminal acetylation decreases the thermostability of ribosomal proteins in exponentially growing yeast cells

Given that Nt-acetylation is important for different protein properties, such as quality control[22,41], protein folding[65], and protein–protein interactions[66,67], we wondered if the mechanism behind the fast degradation of ribosomal proteins in the *naa10Δ* strain was related to defects on protein folding or protein–protein interactions. Thus, we explored structural differences of the *naa10Δ* and WT proteomes under near-physiological conditions by Thermal Proteome Profiling and Data Independent Acquisition (TPP-DIA) (Fig. 6A).

The TPP approach is based on the principle that proteins denature and become insoluble when exposed to heat. By measuring the abundance of proteins in the soluble fraction through a gradient of temperatures, the resulting melting curves reflect protein intra and inter-interactions in the cellular milieu. Changes in protein associations can therefore be inferred through thermal stability readouts in a large-scale manner by mass spectrometry[68,69]. Using this strategy, we determined with high confidence 2689 protein-melting temperatures ($T_m$), defining $T_m$ as the temperature at which 50% of the protein is unfolded (Fig. 6B and Supplementary Data 6).

To identify proteins with changes in their thermal stability between the *naa10Δ* and WT strains, we performed a volcano plot analysis of the protein-melting temperatures (Fig. 6C). This analysis showed that ribosomal proteins of both the large and small ribosome subunits as well as proteins of the proteasome are unstable with lower $T_m$ in the *naa10Δ* condition compared to WT. Contrarily, E3 ubiquitin ligases such as *TOM1*, *CDC16*, *HRT1* and *RSP5* were stabilized with higher $T_m$ in the *naa10Δ*.

KEGG pathway enrichment analysis of proteins with significant changes in thermostability showed that proteins related to the ribosome are destabilized in the *naa10Δ* strain (Fig. 6D). These findings are in agreement with the literature as *TOM1*, *HRT1*, and *RSP5* are ubiquitin ligases related with the degradation of defective and excess ribosomal proteins. For example, *RSP5* is linked to the maintenance of cytosolic ribosome integrity under rich nutrient conditions[70], while *HRT1* and *TOM1* are associated with the degradation of non-functional ribosomes[71,72] and quality control pathway of ribosomes, respectively[59,60].

To validate the results obtained by the DIA-TPP, we performed an isothermal shift assay (ITSA)[73]. The ITSA approach simplifies the DIA-TPP experiment while increasing the statistical power by enhancing the identification sensitivity by quantifying the difference in soluble and precipitated protein fractions at single temperatures (Supplementary Fig. 6A). The individual temperatures were selected according to the melting curves of ribosomal proteins obtained from DIA-TPP ranging from 38.3 to 49.9 °C and divided into four different temperatures (Supplementary Fig. 6B). To visualize proteins showing changes in thermostability, we performed a volcano plot analysis per temperature for the soluble and precipitated fractions, respectively (Supplementary Fig. 6C). As expected, the ribosomal proteins were destabilized in the *naa10Δ* strain when comparing the soluble and precipitated fractions to WT across temperatures. Moreover, since it has been observed that thermal destabilization of cytosolic ribosomes occurs during mitosis in human cells[74], we decided to investigate the cell cycle distribution in the *naa10Δ* vs WT yeast by flow cytometry to discard any bias caused by the enrichment of mitotic cells. The actively growing WT and *naa10Δ* yeast cells showed no significant difference between their cell cycle distribution profiles (Supplementary Figs. 7A, B and 8). Thus, the reduced proliferation rate of the *naa10Δ* strain is caused by an overall delay of the cell cycle progression in general rather than in any particular cell cycle stage. In addition, based on the forward scatter measurements, we were able to corroborate that *naa10Δ* cells were 20% larger than WT cells (Supplementary Fig. 7C), agreeing with what has been reported earlier[7].

Next, we compared the changes in protein turnover with the corresponding melting point differences determined by pSILAC and DIA-TPP, respectively (Fig. 6E). We found that ribosomal proteins exhibiting faster turnover rate, also have a significant shift in their thermostability. In contrast, proteasomal proteins show a significant shift in thermostability but not a significant change in turnover rate. This is consistent with Nt-acetylation of catalytic core proteasomal protein members having been linked to NatB complex with the majority of proteins composing the 20 S proteasome are substrates of that NAT complex. However, it has also been reported that eight subunits of the 19 S proteasome are NatA substrates and that lack of NatA did not result in a significant change in chymotrypsin-like activity of the 26 S proteasome but a higher activity and accumulation level of the catalytic core particle of proteasome 20 S in absence of SDS[61,75,76]. To verify this observation, we plotted the previously confirmed NatA substrates from our N-terminome profiling experiments in the thermostability and turnover rate space (Supplementary Fig. 9A, B). Reassuringly, the global changes in thermostability and turnover rate of NatA substrates resembled what we observed at the protein level. These findings suggest that Nt-acetylation may affect the structure of the 19 S proteasome, which aligns well with our results, suggesting that

Nt-acetylation of ribosomal and proteasomal proteins might be implicated in folding or interaction between proteins of their respective complexes.

Taken together, our data indicate that lack of Naa10 promotes the defective or delayed folding of ribosomal proteins and consequently, increases the probability of their degradation, especially under perturbations such as heat and possibly other stress conditions.

## Lack of NatA-dependent N-terminal acetylation promotes ubiquitination of ribosomal proteins in exponentially growing yeast cells

NatA mutant strains display a pleiotropic phenotype affecting different cellular processes linked to the impairment of protein–protein interactions, transcriptional alterations, and impairment of chaperone systems[7,18,29]. Therefore, it is likely that the UPS system is required to be active to eliminate the damage caused by lack of NatA and maintain cellular proteostasis. According to this hypothesis, the lack of NatA has been related with an increased activity of the UPS system[24,61]. In addition, we found that the ubiquitin ligases involved in the N-degron pathway *TOM1*, *UFD2*, *UFD4*, and *UBR1*[40], as well as proteasomal proteins were upregulated in the *naa10Δ* strain. Consequently, we investigated the role of *NAA10* deletion on protein ubiquitination by using a diGly enrichment approach and DIA-MS (Fig. 7A).

To check if there is a link between fast turnover proteins detected in the pSILAC experiment and UPS system, we overlapped the fast turnover proteins in a volcano plot comparing diGly sites between *naa10Δ* and WT strains (Fig. 7B). As expected, the majority of the fast turnover proteins were more ubiquitinated in the *naa10Δ* strain compared to WT.

As shown previously, the ribosomal proteins were unstable and faster-degraded in the *naa10Δ* strain. To investigate this further, we mapped the proteins of the large and small ribosomal subunits to the differentially expressed ubiquitinome analysis (Fig. 7C and Supplementary Data 7). We found that the ribosomal proteins were more often ubiquitinated in the *naa10Δ* strain. In addition, lysine residues that have been reported not to be accessible in the structure of the mature ribosome and common substrates of *TOM1*[59], as well important ubiquitination events related to RQC (ribosome-associated protein quality control) and NRD (non-functional rRNA decay) pathways such as RPS3-K212[77,78] and RPS20-K8[79] were found in our analysis. This observation suggests that the ubiquitin ligase *TOM1* is active in the *naa10Δ* strain, and further that the ribosomal proteins are actively being ubiquitinated and thereby marked for degradation by the UPS system. To test this hypothesis, we created a double KO strain (*naa10Δ tom1Δ*) and tried to rescue ribosomal proteins from proteasomal degradation (Supplementary Fig. 10A, B). Although only four ribosomal proteins from the large ribosome subunit (RPL7, RPL4, RPL42, and RPL37) were upregulated in the *naa10Δ tom1Δ* compared to *naa10Δ* strain, this result suggests that additional ubiquitin ligases such as UFD2, UFD4 and UBR1 as well as proteasomal proteins, which were consistently upregulated in the different dimensions of the *naa10Δ* proteome, are likely involved in the fast turnover of ribosomal proteins in NatA-lacking cells.

## Discussion

The alteration of the Nt-acetylome results in a pleiotropic phenotype as a consequence of changes in intrinsic properties of the Nt-acetylated proteins, such as lifespan, folding and binding[18]. However, the effect of Nt-acetylation seems to be dependent on the cellular context and the identity of the Nt-acetylated proteins. Our results suggest that the lack of Nt-acetylation carried out by the NatA complex in *Saccharomyces cerevisiae* promotes the fast turnover of a number of NatA substrates. This finding is in agreement with recent studies in other species[17,24,80,81] supporting the concept that across the eukaryotic kingdom, Nt-acetylation increases proteome stability rather than

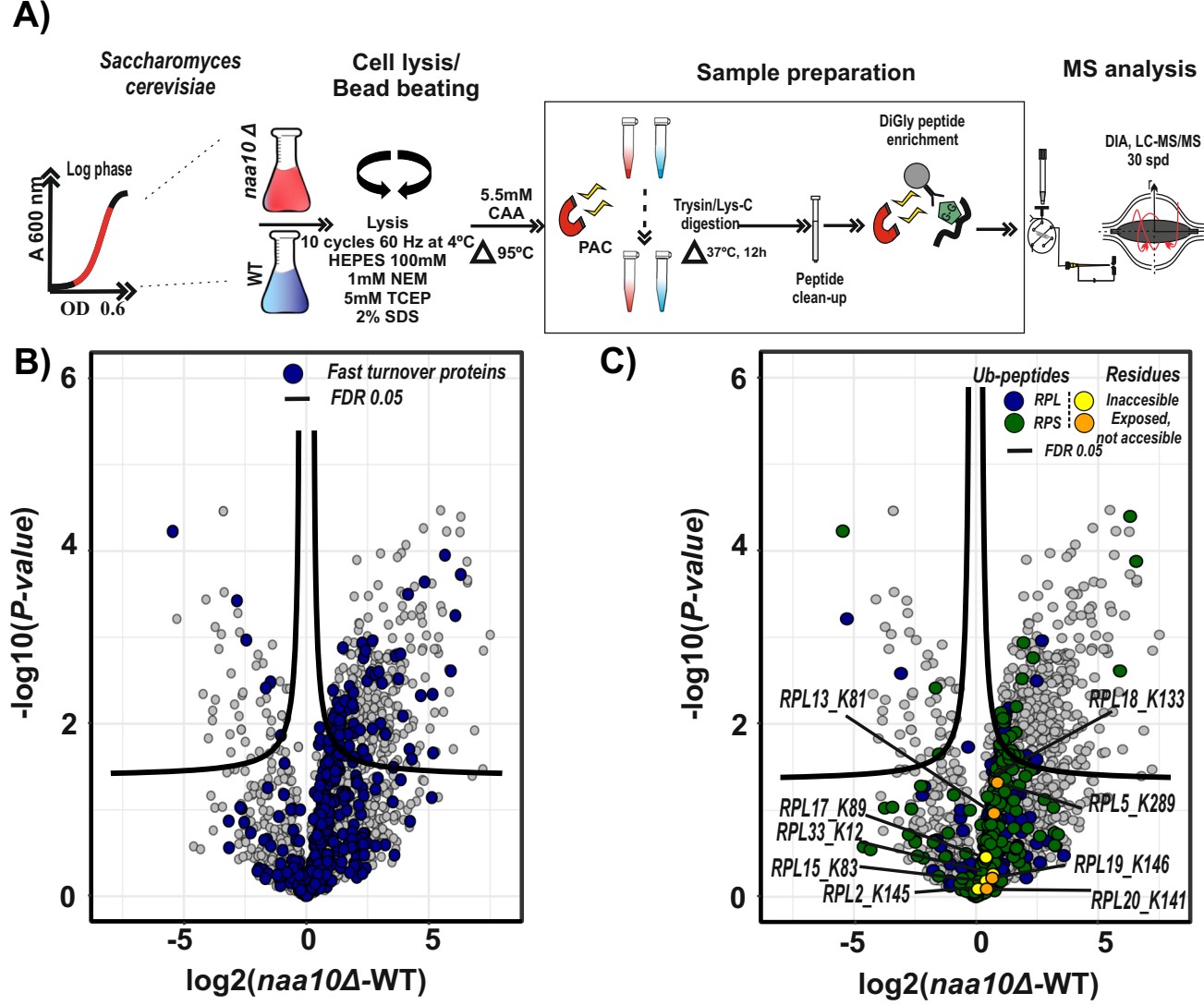

**Fig. 7 | Fast-degraded proteins are ubiquitinated in NatA-defective cells.**
**A** Schematic representation of the implemented ubiquitinome enrichment strategy. Yeast cells grown to mid-log phase were harvested and digested using Lys-C and trypsin prior to peptide clean-up and digly peptide enrichment. Quantification was performed by using label-free intensities (LFQ; label-free quantitation). $n = 3$ replicate cultures per condition. **B** Comparison of the *naa10Δ* and WT systems in a volcano plot to identify ubiquitinated peptide abundance changes (diGly peptides). Significantly regulated proteins at 1% and 5% false discovery rate (FDR) are delimited by dashed and solid lines, respectively (FDR controlled, two-sided *t* test, randomizations = 250, s0 = 0.1). Fast turnover rate proteins determined by pSILAC are highlighted in blue. **C** Same as (**B**) but highlighting large and small subunit ribosomal proteins. Source data are provided as a Source Data file (**B**, **C**).

destabilizing it. In particular, we found that cytosolic ribosomal proteins, which in general are substrates of NatA[53], were consistently affected by the lack of NatA. Briefly, this set of proteins shows a fast turnover and lower thermostability at mid-log phase, as well as a downregulation of some ribosomal proteins at steady state. These results indicate that Nt-acetylation might be involved in protein folding or protein–protein interactions, affecting protein complexes such as the ribosome. On the other hand, proteasomal proteins showed lower thermostability but no change in turnover rate at mid-log phase in the *naa10Δ* strain, suggesting that the lack of Nt-acetylation might affect the proteasome complex but not to a degree, which compromises its stability. This aligns with previous observations arguing that protein degradation requires not just the absence of Nt-acetylation but also other intrinsic features of the target proteins[23]. Noteworthy, the ribosome biogenesis in the *naa10Δ* strain compared to WT showed a normal profile, but increased sensitivity to temperature, as well as protein translation inhibitors such as caffeine and cycloheximide. These results indicate that the Nt-acetylation might contribute to the optimal activity and assembling of the ribosome. In fact, a recent study on pathogenic variants of *NAA15* found these to cause congenital heart disease[82] which could be mechanistically related with our results. Briefly, patient-derived cells expressing pathogenic *NAA15* variants displayed decreased Nt-acetylation of NatA substrates and defects in cardiomyocyte differentiation. Interestingly, the *NAA15*-defective human cells displayed a very specific downregulation of cytosolic ribosomal proteins that could not be mechanistically explained. Given the current data in yeast establishing a functional link between NatA-mediated Nt-acetylation and ribosomal protein stability, the same mechanism is likely to be at play in the cells of humans with heart disease caused by defective NatA. Our results suggest that the lack of NatA promotes the defective or delayed folding of ribosomal proteins or the disruption of their interactions in the full-assembled 80 S ribosome, decreasing the active fraction of ribosomes and increasing their probability of degradation, particularly under stress conditions, which could explain the temperature-sensitive and slow-growth phenotypes described in NatA-lacking cells.

The ubiquitinome analysis revealed the active ubiquitination of the fast-degraded proteins in the NatA-lacking cells. In addition, ubiquitination events on ribosomal proteins from both ribosome subunits and specific ribosomal large subunit sites, which have been described as concealed in the mature ribosomes, suggest that TOM1 is active in the *naa10Δ* strain. Furthermore, we found ubiquitination events associated to NRD and RQC pathways (RPS3-K212[78,79] and RPS20-K8[79], respectively) in the *naa10Δ* ubiquitinome. In contrast, despite observing upregulated autophagy markers such as ATG19 and ATG38 in *naa10Δ* strain, ubiquitination of RPL25-K74[83,84] was observed. This ubiquitination event has previously been reported to prevent the degradation of the 60 S subunit by ribophagy. Therefore, it is likely that the degradation of ribosomal proteins in *naa10Δ* strain occurs mostly through the UPS system. However, the observed impairment of mitochondrial ribosomal protein degradation can be explained by the critical role of NatA in mitophagy[37].

These results potentially link the lack of Nt-acetylation with quality control mechanisms important for regulating multiplex subunit complex and protein folding through the UPS system such as the excess ribosomal protein quality control (ERISQ) pathway[59]. Nevertheless, the upregulation of multiple ubiquitin-conjugating enzymes such as TOM1, UFD2, UFD4, and UBR1 and proteasomal proteins in the NatA-lacking cells suggest that the fast turnover of NatA substrates is the result of a systemic upregulation of the UPS system.

Combined, our results contribute to elucidate the mechanism behind the role of NatA-mediated Nt-acetylation on proteostasis. Specifically, how this modification might be coupled to the protein folding and complex formation as well as the activity of the UPS system, supporting the concept of Nt-acetylation as an avidity enhancer of protein–protein interactions and folding. However, structural and biochemical studies are needed to shed light on the mechanism behind the rules dictating the evolutionarily conserved interactions promoted by Nt-acetylation within the full-assembled ribosome and thus its impact on ribosome function.

## Methods

### Yeast strains and sample preparation

The *S. cerevisiae* strains used were S288C isogenic yeast strains (MATα) wild-type, BY4742 (Y10000, EUROSCARF; internal reference Arnesenlab yTA36); and the thereof modified *naa10Δ*, YHR013C-Δ::kanMX4 (Y10976, EUROSCARF; internal reference Arnesenlab yTA42).

Yeast phenotyping on agar plates were performed as previously[6]. Briefly, yeast strains were grown to the early log phase and serial 1:10 dilutions containing the same number of cells were spotted on agar plates containing various stressors.

For the pSILAC experiment, *S. cerevisiae* strains were grown in synthetic medium containing 6.7 g/l yeast nitrogen base, 2% glucose, 2 g/l dropout mix (Yeast Synthetic Drop-out Medium Supplements without lysine, Y1896 Sigma-Aldrich) containing all amino acids except lysine. For heavy pre-labeling, heavy [$^{13}C_6$/$^{15}N_2$] L-lysine (608041, Sigma-Aldrich) was added to a final concentration of 30 mg/l or 0.436 mM. Triplicate cultures of each strain were precultured three successive times in medium containing heavy lysine overnight at 30 °C. After preculture for SILAC labeling, cells were diluted to $OD_{600} = 0.4$ and cultured in triplicates of 400 ml, still in heavy-Lys medium. After 90 min, cells were transferred to medium containing light lysine (L5501, Sigma-Aldrich), via three rapid and gentle washes in 30 °C preheated medium without lysine at room temperature. At this point $OD_{600}$ was readjusted to 0.4 for all samples. Cells were harvested at given time points by centrifugation (10,000×*g* for 5 min at 4 °C) and $OD_{600}$ was measured for each harvest point. Cell pellets were washed twice with ice-cold water, snap-frozen in liquid $N_2$ and stored at −80 °C. As a control, prior to the transfer from heavy to light lysing medium, 25 ml of heavy-Lys culture was kept and harvested along with the last timepoint.

For all other sample preparations, yeast were grown in complete synthetic medium at 30 °C. Overnight preculture was diluted to $OD_{600} = 0.5$. Cultured yeast were harvested at $OD_{600}$ -1.8 by centrifugation (10,000×*g* for 5 min at 4 °C), washed two times with cold water and stored at −80 °C. For the ubiquitinome analysis, triplicate cultures of yeast WT and *naa10Δ* were grown starting from $OD_{600} = 0.4$ in synthetic complete medium. 0.003% SDS and DMSO were added after 3 h and cells were incubated for further 4 h. Cells were harvested by 10 min centrifugation at 3000×*g* followed by three washes in cold water.

### Yeast flow cytometry with SYTOX Green

Yeast were grown in SC medium and harvested at the same conditions as for the other analyses, before fixation and staining with the DNA-binding dye SYTOX Green (Invitrogen™ #S7020), which has been shown to outperform propidium iodide providing more reliable stain with improved linearity between DNA content and fluorescence[85]. At harvest, 1 ml culture was centrifuged at 4000×*g* for 10 min, before the cells were resuspended in 1.5 ml Milli-Q water. The cells were fixed by drop-wise addition of 3.5 ml 100% ethanol at 1400 rpm vortexing and incubated on a rotating wheel (15 rpm) overnight at 4 °C. The cells were washed in 1 ml Milli-Q water and incubated in 500 μl heat-treated RNase solution (Qiagen #19101) for 4 h at 37 °C. The cells were pelleted, treated with 200 μl pepsin protease solution (Roche #10108057001) for 15 min at 37 °C, before resuspension in 500 μl 50 mM Tris, pH 7.5 and storage at 4 °C. In all, 50 μl cell solution was mixed with 1 ml SYTOX Green solution in a dark microtube and then sonicated at 20 kHz for 5 × 2 s pulses on ice. The DNA content reported by the SYTOX Green signal intensity was measured using Accuri C6 (Flow cytometry core facility, Bergen) and the FL1 detector with a standard 530/30 band pass filter. The limit was set to 5000 cells, and fluidics speed was set to fast. Three independent experiments with seven replica cultures in total were run. Data was processed, analyzed and visualized using FlowJo. Cell cycle analysis was performed using the cell cycle tool in FlowJo applying the Watson (Pragmatic) model and equal range for C1 and C2, optimized for the lowest possible RMSD. Cell cycle distribution % values were exported and statistical testing was performed using two-tailed *t* test with unequal variance.

### Northern blotting analysis of rRNA processing intermediates

Ten μg whole cell RNA from WT and *naa10Δ* strains (biological triplicates) was separated by denaturing gel electrophoresis on a 1% agarose formaldehyde denaturing gel. The RNA was subsequently transferred to a positively charged nylon membrane (BrightStar-Plus, Ambion) by capillary blotting, followed by cross-linking using UV-light. Probes targeting ITS1 (CGGTTTTAATTGTCCTA), ITS2 (TGAGAAGGAAATGACGCT), 18 S (AATTCTCCGCTCTGAGATGG), 5.8 S (GCAATGTGCGTTCAAAGA), and 25 S (GATCAGACAGCCGCAAAAAC), 10 pmol each, were labeled with [γ-32-P]-ATP using T4 polynucleotide kinase (Thermo Fisher Scientific) and hybridized to the membrane in hybridization buffer (4× Denhardts solution, 6× SSC, and 0.1% SDS), at 45 °C for 16 h. Subsequently, the membranes were washed four times in washing buffer (3× SSC and 0.1% SDS) and then exposed to a Phosphor Imager (PI) screen overnight for the ITS1 and ITS2 probe and 10 minutes for the 18 S, 5.8 S, and 28 S probes to avoid saturation. The PI screens were scanned using a Typhoon scanner (GE Healthcare) and analyzed by Fiji-ImageJ software.

### Preparation of samples for LC−MS/MS analysis

For global proteome profiling and pSILAC, yeast cells were resuspended 1:2 in lysis buffer composed of 100 mM Tris(hydroxymethyl) aminomethane (Tris), pH 8.5, 5 mM, Tris(2-carboxyethyl)phosphine hydrochloride (TCEP), 10 mM chloroacetamide (CAA) and 2% sodium dodecyl sulfate (SDS). Cells were lysed by eight rounds of bead beating (1 min beating, 1 min rest, 66 Hz) in a Precellys 24 homogenizer with

400 μm silica beads (2:1, resuspended cells: silica beads). The extracted protein lysates were heated to 95 °C during 10 min, briefly sonicated and centrifuged at 16,000×*g*, 4 °C. Afterwards, the protein concentration was approximated using the BCA assay (Pierce™). The resulting samples were digested overnight using the protein aggregation capture (PAC) protocol[86]. The proteolytic digestion for pSILAC was performed by addition of lysyl endopeptidase (Lys-C, Wako), 1:50 enzyme to protein ratio, and incubated at room temperature overnight. For all other sample preparations, lysyl endopeptidase (Lys-C, Wako) and trypsin (Tryp, Sigma-Aldrich) were added 1:300 and 1:100 enzyme to protein ratio, respectively and incubated at 37 °C overnight. The digestion was quenched by the addition of trifluoroacetic acid (TFA, Sigma-Aldrich) to final concentration of 1%. The resulting peptide mixtures were desalted and stored on Sep-Pak columns (Waters) at 4 °C until further use. Three independent cultures by condition were analyzed.

Offline High pH Reversed-Phase HPLC Fractionation.100 μg of peptides were separated by HpH reversed-phase chromatography using a Waters XBridge BEH130 C18 3.5 μm 4.6 × 250 mm column on an Ultimate 3000 high-pressure liquid chromatography (HPLC) system (Dionex, Sunnyvale, CA, USA) operating at a flow rate of 1 ml/min with three buffer lines. Buffer A H₂O, Buffer B C₂H₃N (ACN) and Buffer C 25 mM NH₄HCO₃, pH 8 (Ammonium bicarbonate). The separation was performed by a linear gradient from 5% B to 35% B in 62 min followed by a linear increase to 60% B in 5 min, and ramped to 70% B in 3 min. Buffer C was constantly introduced throughout the gradient at 10%. A total of 12 fractions were collected at 60 s intervals. Samples were acidified after digestion to final concentration of 1% trifluoroacetic acid (TFA). In total, 250 ng of each sample were loaded into Evotips (Evosep) for LC−MS/MS analysis.

TPP-DIA and ITSA LC−MS/MS analysis. Yeast pellets were resuspended in 1 ml lysis buffer1 composed of 0.1% nonyl phenoxypolyethoxylethanol (NP-40) and Phosphate-Buffered Saline (PBS, 137 mM NaCl, 2.7 mM KCl, 10 mM Na₂HPO₄ and KH₂PO₄, Sigma-Aldrich) supplemented with protease inhibitor (cOmplete™ Protease Inhibitor Cocktail, Roche) at room temperature (RT). In all, 400-μm silica beads were added to 200 μl of the resuspended cells 2:1 suspension to beads ratio. The cells were lysed by eight rounds of bead beating (1 min beating, 3 min rest, 66 Hz) at 4 °C. The protein concentration was approximated using the BCA assay (Pierce™). In total, 100 μl of lysate at 2 μg/μl of each sample were transferred to a 96-well plate and keep to room temperature for 10 min. The samples were transferred to specific wells according to the desired temperature. The cell lysates were heated at their respective temperatures in a thermocycler for 4 min and immediately incubated at RT for an additional 4 min. The resulting cell lysates were transferred to a 1.5 ml microcentrifuge tubes and centrifuged at 20,000×*g*, 4 °C for 1 h. 50 μL of the supernatant were transferred to a deep well plate. Afterward, 50 μl of lysis buffer2 composed of 200 mM Tris, pH 8.5, 10 mM TCEP, 20 mM CAA and 4% SDS. The resulting protein lysates were heated at 95 °C for 10 min. The samples were digested overnight by a 96-well format automatized PAC[86] workflow optimized for the KingFisher™ Flex robot (Thermo Fisher Scientific). Briefly, the 96-well comb was stored in plate #1, the sample in plate #2 in a final concentration of 70% acetonitrile and with 50 μl of magnetic Amine beads (ReSyn Biosciences) in a protein-to-bead ratio of 1:2. Protein aggregation capture was performed in two steps of 1 min mixing and 10 min pauses. The sequential washes were executed in 2.5 min. Washing solutions are in plates #3–5 (95% Acetonitrile (ACN)) and plates #6–7 (70% Ethanol). Plate #8 contains 300 μl digestion solution of 50 mM ammonium bicarbonate (ABC), 0.5 μg of Lys-C (Wako) and 1 μg trypsin (Sigma-Aldrich). The digestion was quenched by the addition of trifluoroacetic acid (TFA, Sigma-Aldrich) to a final concentration of 1%. Six independent cultures per condition were analyzed.

For the precipitate analysis by ITSA, the remaining supernatant after centrifugation was discarded, and 100 μl fresh lysis buffer1 were added to the 1.5-ml microcentrifuge tubes. The resulting suspension was centrifuged at 20,000×*g*, 4 °C for 1 h. This operation was repeated two times. Afterward, the supernatant was discarded, and the precipitate was solubilized with lysis buffer and heated at 95 °C during 10 min. The samples were digested overnight by a 96-well format automatized PAC[86] as stated previously.

For the TTP-DIA experiment, the peptide concentration for the two lowest temperatures was spectrophotometrically determined at 280 nm using a NanoDrop instrument (Thermo Scientific) and the average used as the concentration for all samples. For ITSA, the peptide concentration for each sample was determined. An equivalent of 500 ng of each sample were loaded into Evotips (Evosep) for LC−MS/MS analysis.

N-terminal-enrichment. The enrichment was performed as outlined previously[87] with some modifications. Briefly, the yeast cells were lysed, and the protein concentration quantified as in yeast proteome profiling experiment. Afterward, magnetic SiMAG-Sulfon beads (Chemicell, 1202) were added to protein lysate to beat ratio of 1:10 ratio (w/w). Pure 100% ACN was added to a final volume 70% v/v to initiate binding. After 10 min incubation at RT, the mixture was kindly vortexed and incubated for additional 10 min. The supernatant was removed with help of a magnet and the beads were rinsed with 1 ml ACN and 1 ml 70% ethanol. Beads were resuspended in 100 μl 100 mM HEPES, pH 8. 2 M folmadehyde (Sigma-Aldrich) and 1 M sodium cyanoborohydride (Sigma-Aldrich) were added to a final concentration of 30 mM and 15 mM, respectively. The lysate was incubated at 37 °C for 1 h. Following this, fresh labeling reagents were added, and the lysate was incubated for an additional hour. The reaction was quenched by the addition of 4 M Tris, pH 6.8 to a final concentration of 500 mM and incubated for 3 h. Additional magnetic beads were added at a 1:5 ratio and protein bound by addition of 100% ACN to a final concentration of 70% v/v. Beads were settled on a magnetic stand after 20 min incubation at RT and vortexed each 10 min. The supernatant was removed, and the beads rinsed with 1 mL ACN and 1 ml 70% ethanol. The beads were resuspended in 300 μl of 200 mM HEPES buffer, pH 8.0. The digestion was performed with trypsin (Tryp, Sigma-Aldrich) 1:100 enzyme to protein ratio and incubated 37 °C overnight. After the digestion, 100% ethanol was added to the proteome digest to a final concentration of 40% v/v before the addition of undecanal (EMD Millipore) at an undecanal to peptide ratio of 20:1 w/w and addition of 1 M sodium cyanoborohydride to a final concentration of 30 mM. The mixture was incubated at 37 °C for 1 h. Afterwards, the mixture was bounded to a magnetic rack and the supernatant transferred to a low-binding tube. The supernatant was acidified to pH=2 with 0.1% TFA in 40% ethanol to a loading volume of 500 μl and loaded on a Sep-Pak column (Waters). The Sep-Pak columns were conditioned with 1 ml methanol followed by 3 ml 0.1% TFA in 40% ethanol. The flow-through was collected in 1.5-ml protein Low-bind tubes and concentrated by vacuum centrifugation. Finally, the resulting peptides were resuspended in 0.1% TFA and desalted using Sep-Pak columns (Waters). The resulting peptides were storage using Sep-Pak columns (Waters) until use at 4 °C. Four independent cultures per condition were measured.

DiGly peptide enrichment. Yeast cells were resuspended 1:2 in lysis buffer composed of 100 mM Tris, pH 8.5, 5 mM, TCEP, 1 mM N-ethylmaleimide (NEM) and 2% SDS. Cells were lysed by bead beating and the resulting lysate was supplemented with CAA to a final concentration of 5.5 mM. DiGly peptide enrichment was performed using the PTMScan® Ubiquitin Remnant Motif (K-ε-GG) Kit (Cell Signaling Technology (CST)). Briefly, 1 mg peptides per sample were reconstituted in 1.5 ml PTMScan HS IAP Bind Buffer and 20 μl of cross-linked antibody magnetic beads were added to each sample tube. The tubes were incubated on an end-over-end rotator for 2 h at 4 °C. Afterward, the tubes were spun at 2000×*g* for 5 s and placed in a magnetic rack for 10 s. The supernatant was discarded and 1 ml HS IAP Wash Buffer was mix with the beads. The tubes were placed again in a magnetic rack and

the supernatant discarded. This operation was repeated four times. Subsequently, the tubes were washed with LC–MS water two times. Finally, the DiGLY peptides were eluted form the magnetic beads by adding 50 μl of IAP Elution Buffer (0.15% TFA) to the beads for 10 min at room temperature and 500 rpm. Tube was placed in a magnetic rack and the supernatant was transferred to a new microcentrifuge tube. This operation was repeated two times. The peptide concentration was spectrophotometrically determined at 280 nm using a NanoDrop instrument (Thermo Scientific), and the equivalent of 500 ng of each sample were loaded onto Evotips (Evosep) for LC−MS/MS analysis. Three independent cultures per conditioned were analyzed.

## LC−MS/MS analysis

All samples except those with Nt-enrichment were analyzed on the Evosep One system (Evosep) using a 15 cm, in-house packed, reversed-phase column (150 μm inner diameter, ReproSil-Pur C18-AQ 1.9 μm resin [Dr. Maisch GmbH]). The column temperature was controlled at 60 °C using a using an integrated column oven (PRSO-V1, Sonation, Biberach, Germany) and binary buffer system, consisting of buffer A (0.1% formic acid (FA), 5% ACN) and buffer B (100% ACN) and inter-faced online with the Orbitrap Exploris 480 MS (Thermo Fisher Scientific, Bremen, Germany) using Xcalibur (tune version 3.0). pSILAC and proteome profiling experiment was measured with the pre-programmed gradient for 60 samples per day (SPD). For all other experiments, the pre-programed gradient correspond to 30 SPD was used.

Nt-enrichment was analyzed in an EASY-nLC 1200 system (Thermo Fisher Scientific), using a 15 cm, in-house packed, coupled online with the Orbitrap Q Exactive HF-X MS (Thermo Fisher Scientific, Bremen, Germany), nanoflow liquid chromatography, at a flow rate of 250 nl/min. The total gradient was 60 min followed by a 17 min wash-out and re-equilibration. Briefly, the flow rate started at 250 nl/min and 8% ACN with a linear increase to 24% ACN over 50 min followed by 10 min linear increase to 36% ACN. The washout flow rate was set to 500 nl/min at 64% ACN for 7 min followed by re-equilibration with a 5 min linear gradient back down to 4% ACN. The flow rate was set to 250 nl/min for the last 5 min.

For the DDA experiments. The Orbitrap Q Exactive HF-X MS was operated in Top6 mode with a full scan range of 375–1500 $m/z$ at a resolution of 60,000. The automatic gain control (AGC) was set to 3e6 with a maximum injection time (IT) of 25 ms. Precursor ion selection width was kept at 1.4 $m/z$ and peptide fragmentation was achieved by higher-energy collisional dissociation (HCD) (NCE 28%). Fragment ion scans were recorded at a resolution of 30,000, an AGC of 1e5 and a maximum fill time of 54 ms. Dynamic exclusion was enabled and set to 30 s. The Orbitrap Exploris 480 MS was operated in Top12 mode with a full scan range of 350–1400 $m/z$ at a resolution of 60,000. AGC was set to 300 at a maximum IT of 25 ms. Precursor ion selection width was kept at 1.3 $m/z$ and fragmentation was achieved by HCD (NCE 30%). Fragment ion scans were recorded at a resolution of 15,000. Dynamic exclusion was enabled and set to 30 s.

For the DIA experiments. The Orbitrap Exploris 480 MS was operated at a full MS resolution of 120,000 at $m/z$ 200 with a full scan range of 350–1400 $m/z$. The full MS AGC was set to 300 with an IT 45 ms. Fragment ion scans were recorded at a resolution of 15,000 and IT of 22 ms. In all, 49 windows of 13.7 $m/z$ scanning from 361 to 1033 $m/z$ were used with an overlap of 1 Th. Fragmentation was achieved by HCD (NCE 27%).

## Raw MS data analysis

For publication, all the raw files corresponding to pSILAC, Nt-enrichment and pSILAC-6h were analyzed with MaxQuant (1.6.7.0) and searched against a UniProt's yeast protein sequence database as follows. For pSILAC, a database composed of the canonical isoforms of *S. cerevisiae* proteins as downloaded from UniProt in 2019, which was

customized by removing all signal peptides as annotated in UniProt. For Nt-enrichment and pSILAC-6 h, a data database composed of the canonical isoforms of *S. cerevisiae* proteins, as downloaded from Uni-Prot in 2019 was used (https://www.uniprot.org/). For pSILAC analysis, the multiplicity was set to two allowing the detection of light (K0) and heavy (K8)-labeled peptides. Cysteine carbamylation was set as a fixed modification, whereas methionine oxidation and protein N-termini acetylation were set as variable modifications. Match between runs (MBR) was enabled. For Nt-enrichment and pSILAC-6h the default settings were kept, and MBR was disabled.

TPP-DIA, ITSA, and Ubiquitinome raw files were analyzed using Spectronaut v15 (Biognosys) with a library-free approach (directDIA) using a database composed of the canonical isoforms of *Saccharomyces cerevisiae* proteins as downloaded from UniProt in 2019, which was customized by removing all signal peptides as annotated in Uni-Prot. This customized data based was supplemented with a common contaminant database. For TPP-DIA and ITSA, cysteine carbamylation was set as a fixed modification, whereas methionine oxidation and protein N-termini acetylation were set as variable modifications. For the Ubiquitinome experiment, DiGLY (K,T,S) was defined as an additional variable modification and PTM localization was enabled and set to 0.5. Precursor filtering was set as Q-value, and cross-run normalization was turned off for TPP-DIA. ITSA analysis was performed with Spectronaut using default settings. For ubiquitination, the imputation setting was disabled. Further processing analyses were performed either in R (v4.2.1), Prostar (v1.28.0)[88] or Perseus (v1.6.7.0).

## Bioinformatic data analysis

For all analyses, common contaminants and proteins hitting the reverse decoy database were filtered out prior to analysis. The pSILAC experimental data analysis was performed using R (v4.2.1). Briefly, all protein identifications identified with less than two unique peptides and detected in less than four time points were discarded. The heavy-label incorporation was calculated from MaxQuant heavy-to-light intensity ratios. The relative isotope abundance (RIA) was calculated as follows:

$$RIA_t = ratio\ H/L / (1 + ratio\ H/L) \tag{1}$$

As the RIAt collected in the time domain follows an exponential curve of the form:

$$RIA_t = Ae^{-kt} \tag{2}$$

The exponential curve was linearized for the derivation of protein turnover parameters, using RIAt values up to 6 h. The linearization was performed by taking the natural logarithm of both sides of the equation and rearranging:

$$\ln RIA_t = -kt + \ln A \tag{3}$$

By comparing the Eq. (3), to a linear model y = mx + b, the slope corresponds to $k$ and the intercept corresponds to ln A. We calculate the half-life (t½) of each protein as the time when the protein is half-labeled (i.e., RIA = 0.5). Thus, t ½ was calculated as follows:

$$t\,1/2 = \ln(2)/|k| \tag{4}$$

For N-terminal peptide analysis from the pSILAC experiment, all identified peptides were filtered to all keep those covering the first or second amino acid of annotated protein sequences retrieved from the UniProt database. Afterward, the identified peptides were grouped by condition and the protein turnover parameters were determined as mentioned previously. Afterward, the median absolute deviation (MAD) of the calculated Kdil condition and replicate was calculated.

Kdil values outside the MAD range relative to the median were deemed as outliers.

TPP-DIA experiment was performed in R (v4.2.1). Briefly, data was grouped by condition and treatment prior log2 transformation. For normalization, to equal out differences in samples that result from unequal sample concentration, normalization was performed using VSN approach (Variance stabilizing Optimization) from the VSN package (v3.15) and implemented in the Prostar pipeline[88]. Afterward, all protein identifications per condition and replicate were treated separately. Low-intensity values that were not part of distribution and not valid values were filtered out. Only proteins with at least eight data points were used for fitting. For fitting the melting curve trajectories of each protein, a four-parameter logistic curve model was used as follows:

$$[Protein] = d + \frac{a - d}{1 + (\frac{T}{T50})^{-d}} \quad (5)$$

Where:

T = temperature

[Protein] = Protein intensity

a = estimated [Protein] at minimum value of T

d = estimated [Protein] at maximum value of T

$T_{50}$ = mid-range T.

b = slope at the inflection point.

Afterward, the median absolute deviation (MAD) of the calculated $T_{50}$ per protein and condition was calculated. $T_{50}$ values outside the MAD range relative to the median were set as outliers.

For calculating the maximum number of protein-melting temperatures and half-lives, the identified proteins were grouped by condition, and the $T_{50}$ and $T_{1/2}$ were calculated as stated above. For statistical analysis, each replicate were treated separately.

For the ITSA, the protein groups table output from Spectronaut v15 was analyzed. The lowest temperature of each condition (25 °C) was used to calculate the soluble and precipitated fraction for different conditions. For ubiquitinome, the analysis was performed as previously[89]. Briefly, DiGLY values were filtered to contain >50% valid values in at least one experimental condition. Missing values were imputed based on a normal distribution width and downshift of 1.8 and a width of 0.3.

The GO term annotation was performed using the R packages: GO.db (v3.8.2) and the genome-wide annotation for Yeast database package Org.Sc.sgd.sd (v3.8.2). The gene set enrichment analysis using KEGG terms was performed with the function gseKEGG from the Clusterprofiler R package (v3.15). The icelogo plots were built using the iceLogo web tool found (https://iomics.ugent.be/icelogoserver/). The statistical analyses were conducted using the Krustal–Wallis, Kolmogorov–Smirnov and Wilcoxon test. The *P* values were corrected according to Benjamin–Hochberg. For volcano plots, *P* values were calculated by unpaired two-tailed Student's *t* test. Statistical significance is indicated in the figure legends.

### Reporting summary

Further information on research design is available in the Nature Portfolio Reporting Summary linked to this article.

## Data availability

The mass spectrometry-based proteomics data generated to determine pSILAC chase, TPP-DIA, ITSA, Deep proteome, and ubiquitinome at steady-state analysis in this study have been deposited in the ProteomeXchange Consortium via PRIDE with the identifier PXD037510. Processed pSILAC chase, TPP-DIA, ITSA, Deep proteome, and ubiquitinome at steady-state analysis data are provided in the Supplementary Information/Source Data file. The Flow cytometry data generated in this study have been deposited on flowrepository.org under the identifier FR-FCM-Z6KA [http://flowrepository.org/id/FR-FCM-Z6KA]. Source data are provided with this paper.

## Code availability

The complete R script code used to perform the analyses is available from the corresponding authors on request. Half-live and Melting curves calculation curve code is provided as Supplementary Data 8.

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

## Acknowledgements

Work at the Novo Nordisk Foundation Center for Protein Research is funded in part by a generous donation from the Novo Nordisk Foundation (grant NNF14CC0001) and the European Union's Horizon 2020 research and innovation program (grant EPIC-XS-823839). U.H.G. was supported by the Novo Nordisk Foundation's Copenhagen Bioscience PhD Program (grant NNF16CC0020906). This work was supported by grants from the Swedish Research Council (Grant 2021-04655 to M.E.J.), the Norwegian Health Authorities of Western Norway (F-12540 to T.A.), the Norwegian Cancer Society (171752-PR-2009-0222 to T.A.), the European Research Council (ERC) under the European Union Horizon 2020 Research and Innovation Program (Grant 772039 to T.A.). The flow cytometry analysis was performed at the Flow Cytometry Core Facility, Department of Clinical Science, University of Bergen, Norway.

## Author contributions

Conceptualization: J.V.O., M.E.J., U.H.G. and T.A. Methodology: H.A., U.H.G., R.R., and N.K. Investigation: U.H.G. Visualization: U.H.G. Supervision: J.V.O., L.J.J., M.E.J., and T.A. Writing—original draft: U.H.G. Writing —review and editing J.V.O., T.A., L.J.J., M.E.J., R.R., N.K., and H.A.

## Competing interests

The authors declare no competing interests.
