## [Peer Review File · Nature Communications]

REVIEWER COMMENTS

Reviewer #1 (Remarks to the Author):

Guzman et al systematically study the effect of the loss of N-terminal acetyltransferase A activity on protein abundance, turnover, thermal stability and ubiquitination states. The biological question is interesting and the results obtained novel. The technological approaches are top notch and well executed. I am in general positive towards publication with some comments for improvement.

Comments:

Cytosolic ribosomes have been observed to be destabilized before in a cell cycle dependent manner in human cells (PMID: 29706546), in mitosis. The authors could consider comparing the distribution of cell cycle stages between WT and deletion mutant to see if enrichment of mitotic cells could partially explain the observation.

Since the authors extensively profile the N-terminome in both conditions and have a good idea of which are the substrates of NatA it would be worth to specifically represent the turnover and thermal stability changes of this group of proteins to better grasp the global changes.

Reference 69 should be the original TPP paper: PMID: 25278616

Reviewer #2 (Remarks to the Author):

The manuscript entitled 'Loss of N-terminal acetyltransferase A activity induces thermally unstable ribosomal proteins and increases their turnover' by Guzman and coworkers describes the impact of absent N-terminal acetylation (NTA) of nascent chains on global protein abundance, thermal stability, and ubiquitination rates in the unicellular model eukaryote yeast.

The study is well designed and considers that N-terminal acetyltransferase A (NatA) depleted yeast mutants grow slower than the wild type affecting the dilution of proteins by cell division. This is a unique selling point of this study and allows a better understanding of the physiological consequences of absent NTA on NatA substrates. The authors combined state-of-the-art global proteomics approaches addressing proteome abundance, NTA frequency, turnover rates (pSILAC), thermal stability (TPP-DIA combined with ITSA), and the ubiquitinome (DiGly peptide enrichment) to dissect the direct impact of lowered NatA substrate acetylation from secondary effects at the global scale. Such secondary effects cannot be avoided by the study design since the ribosome-tethered NatA addresses approximately 40% of the yeast proteome.

In contrast to previous studies in yeast, the authors could demonstrate a stabilizing impact of NTA on NatA substrates. These findings are adequately discussed within the context of previous work in yeast and other eukaryotes. Furthermore, the authors link the absent acetylation of NatA substrates to stronger ubiquitination and faster protein degradation. This aspect could even be strengthened in the manuscript (see suggestions/minor concerns). In the case of ribosomal NatA substrates, absent acetylation decreases the thermal stability of these proteins, which might be causative for the observed increased turnover. Remarkably, the enhanced turnover of NatA substrates was not accompanied by decreased protein abundance, which has also been shown for selected NatA substrates in other eukaryotes. However, the obtained findings are novel and far from being of confirmative nature since they uncover a so-far unknown detailed view of the protein fate at the global scale in response to NTA.

The authors provided compelling evidence for their claims. This reviewer did not spot any flaws in the experimental planning or the interpretation of the large data sets.

Suggestions / minor concerns

Title: The title should specify the organism used in the study since it is currently debated if this mode of action is conserved in other opisthokonts.

Page 5: This analysis revealed that the free Nt-peptides have a significantly faster turnover rate compared to the Nt-acetylated peptides... -> A separate supplemental file listing the corresponding proteins and their predicted/known subcellular localization would be helpful for interested readers who are not proteomics experts.

Page 2: , drought stress -> replace reference 7 with reference 32

Page 10 ...As expected, the majority of the fast turnover proteins were more ubiquitinated in the naa10delta strain... -> The authors should provide information on the identity of proteins and their subcellular localization (e.g., membrane association) in a separate supplemental file and shortly discuss what this means for degradation of the proteome via autophagy or the UPS. Potentially, recent findings by Shen and coworkers (Cell Reports, 2021) on autophagy induction in yeast by absent NatB might be discussed.

Page 10 with recent studies in other species -> add Gong et al (Mol. Plant 2022)

Page 10, as well as protein inhibitors -> protein translation inhibitors

Reviewer #3 (Remarks to the Author):

This manuscript by Guzman et al. studies the functional impacts of N-acetylation on the protein stability in yeast by the proteomics approaches. To investigate the effect of lacking N-acetylation on a proteome-wide scale, authors employ the *naa10Δ* strain, which leads to abolishing the NatA activity, for all proteomic analyses. The pulsed-SILAC analysis shows that the degradation of ribosomal proteins is increased in the *naa10Δ* strain and the thermal proteome profiling reveals the reduction of thermostability of ribosome proteins, suggesting that lack of Naa10 promotes the defective or delayed folding of ribosomal proteins, which increases the probability of their degradation.

The proteomic analyses are well performed and confirmed previous observations. However, the weak point of this paper is that the results presented in this paper provide just the resource of the proteome in *naa10Δ* strain, and there is no functional analysis of the N-acetylation of ribosomal proteins.

The authors focus on the stability of the ribosomal proteins and claim that the lack of Naa10 promotes the defective or delayed folding of ribosomal proteins and consequently, increases the probability of their degradation. However, to address the functional role of N-acetylation of ribosome proteins, the authors should have addressed how N-acetylation is involved in the folding of the ribosomal proteins. For example, many ribosomal proteins have their unique chaperone, which helps the protein folding, the import into the nucleus, and the assembly into pre-mature ribosome particles and so on. Is the N-acetylation involved in such chaperone function?

The authors should have at least investigated and discussed the role of N-acetylation on the fate of ribosome proteins from co-translational folding to assembly step into pre-mature ribosome; northern blotting of premature rRNA is not enough to discuss the effects on the ribosome biogenesis. When and where the ribosomal proteins are degraded? co-translational ? before or after import into nucleolus ? or after assembled into ribosome ?

The authors claim that the *naa10Δ* strain shows increased sensitivity to the protein translation elongation inhibitor cycloheximide and caffeine (Fig. 4D). However, the cell growth of both wild-type and *naa10Δ* are impaired by the addition of cycloheximide and caffeine with the same extent; there is no difference between wild-type and *naa10Δ* strain.

Reviewer #4 (Remarks to the Author):

In this manuscript Guzman et al. present a series of high-quality proteomics experiments to study the role of the main eukaryotic enzyme, N-terminal acetyltransferase (NatA), that mediates protein N-terminal acetylation in *Saccharomyces cerevisiae*. The sounded results allowed the authors to conclude that in yeast NatA-dependent N-terminal acetylation plays a role in protein stability and protein quality control in order to maintain cellular proteostasis, as it has been suggested already in plants and mammalian cells.

The authors started with a global and almost completed protein expression profiling comparing WT and Naa10^Δ (NatA KO) strain proteomes and found significant differences between the two proteomes, specifically ribosomal proteins increased in the KO strain. Then, they performed a pulse SILAC experiment to compare half-lives of proteins between WT and Naa10^Δ observing a general increased turnover of proteins, also ribosomal, in the NatA KO strain implicating thus Nt-acetylation in protein stabilization. With thermal proteome profiling and isothermal shift assays, the authors further connected the increased ribosomal protein turnover to decreased protein stability suggesting defects in folding or interactions.

Finally, by using enrichment of ubiquitinated peptides combined with quantitative mass spectrometry the authors found lack of NatA in yeast leads to an increased ubiquitination of most proteins, and particular ribosomal proteins, which would lead to an increased degradation via the proteasome, explaining thus the divergence of the proteomes between wt and NatA KO strains.

In summary, this study revealed that in yeast NatA-mediated Nt acetylation in yeast is critical for protein stabilization, including ribosomal proteins, most likely by promoting optimal folding and/or supporting protein-protein interactions by avidity.

These findings are of great general interest and represent an important contribution to the understanding of how organism control cellular proteostasis beyond state of the art which merits publication in Nature Communications. Conclusions are drawn by sounded experimental data and robust statistical analysis. Overall, this is a well written manuscript with data and methodology aligning with recent advances in the field. I therefore recommend publication of this manuscript with very minor revisions, see points below.

Minor points:

Fig. 1SB: Under the third and fourth column, there is written Naa20 and should be Naa15. There is no indication about which statistical test and p-value has been performed in order to get the significant differences between WT and Naa10 Δ . Significant differences should be indicated as well in the figure.

Fig. 1

The abbreviation RP, RPL and RPS are quite intuitive but should be also spelt, for example in the legend of Figure 1.

Page 4 last line first paragraph is referenced to Fig. S1C, but should be Fig. S1B

Page 5, second paragraph a bracket around Fig. 2C is missing.

Figure 2: please mentioned in the figure legend what Tdil is.

Figure 3:

(A) In the manuscript text the data of this panel is clear but the graph in the figure is not easy to interpret. Please, revise graph and figure legend and make it clear.

Also, the legend indicates twice that the Tcc is marked by an arrow, but this is not clearly visible. Please, revise this as well.

(B) This figure has been done for the whole proteome. How does the curves look focusing only on NatA targets? Is the shift of the curve between WT and Naa10 Δ more pronounced? This would be an interesting point to add to the manuscript.

(C+D) The comparisons of the whole proteome set to the whole N-terminome dataset does not seem meaningful. Here, the individual protein overlap between the proteome and Nt-peptides is not stated. Would one not expect that if the protein is degraded then corresponding Nt-peptide is also degraded? The data of protein and N-termini should go hand in hand? Thus, it is not clear why to expect significant differences between proteome and Nt-terminome since the Nt-terminome should represent a quite random subset of the proteome. It is as well not indicated how many peptides are Nt-acetylated or free. I would suggest to divide the N-terminome in potential NatA targets (eventually including Naa50 targets as they might be affected as well) and targets of NatB and C and

then compare the proteome data. Alternative, divide the proteome as well in subgroups based on NatA targets and/or other Nats.

Ref 51 is not upper case.

Fig 6D: it is not clear what the grey box with NA means.

Page 9, paragraph 3 there is a typo "catalityncore".

Point-by-point rebuttal letter to REVIEWER COMMENTS

Manuscript: NCOMMS-22-44688

“Loss of N-terminal acetyltransferase A activity induces thermally unstable ribosomal proteins and increases their turnover” by Guzman et al.

Our responses to the reviewers' comments are provided below in blue font. Key modifications and updates are indicated by using yellow highlighted text in the revised manuscript file.

General response: We would like to thank all of the reviewers for their constructive criticism and useful comments, which we believe have helped us to improve our manuscript. Please find answers to the specific points raised by the reviewers below.

Reviewer #1 (Remarks to the Author):

Guzman et al systematically study the effect of the loss of N-terminal acetyltransferase A activity on protein abundance, turnover, thermal stability and ubiquitination states. The biological question is interesting and the results obtained novel. The technological approaches are top notch and well executed. I am in general positive towards publication with some comments for improvement.

Comments:

1.-Cytosolic ribosomes have been observed to be destabilized before in a cell cycle dependent manner in human cells (PMID: 29706546), in mitosis. The authors could consider comparing the distribution of cell cycle stages between WT and deletion mutant to see if enrichment of mitotic cells could partially explain the observation.

We thank the reviewer for pointing out that there could be a correlation between enrichment of mitotic cells and the observed effect on ribosomal proteins in the deletion mutant compared to WT. Thus, we implemented the methodology described in (PMID: 29706546) and investigated the cell cycle distribution in the *naa10Δ* vs WT yeast by flow cytometry. Briefly, yeast cells (*naa10Δ* vs WT) were grown in SC medium (Synthetic complete medium) and harvested at the same conditions as the TPP experiment. Afterwards, yeast cells were fixed and stained with SYTOX Green to measure the cell cycle based on DNA content. Reassuringly, the actively growing WT and *naa10Δ* yeast showed similar cell cycle distribution profiles (Fig. S7A) and cell cycle analysis showed no statistically significant difference (Fig. S7B). Thus, the reduced proliferation rate of the *naa10Δ* strain is caused by an overall delay of the cell cycle progression in general rather than in any particular cell cycle stage.

It should be mentioned that a minor and non-significant increase in G2 cells was observed in these data, however it is unlikely to explain the destabilization of cytosolic ribosomes found in our analyses, since the potential enrichment of mitotic cells in the *naa10Δ* yeast

is minor ($2\% \pm 2.6\%$ potential increase). Based on the forward scatter, this flow analysis also provided information about cell size, showing that *naa10Δ* cells were 20% larger than WT cells (Fig. S7C), agreeing with what has been reported earlier (PMID: 33535049).

These results were summarized in the supplementary figure showed below (Fig. S7) and described in the results section, lines (387-397), which now reads:

Moreover, since it has been observed that thermal destabilization of cytosolic ribosomes occurs during mitosis in human cells (PMID: 29706546), we decided to investigate the cell cycle distribution in the *naa10Δ* vs WT yeast by flow cytometry to discard any bias caused by the enrichment of mitotic cells. The actively growing WT and *naa10Δ* yeast cells showed no significant difference between their cell cycle distribution profiles (Fig. S7A-B). Thus, the reduced proliferation rate of the *naa10Δ* strain is caused by an overall delay of the cell cycle progression in general rather than in any particular cell cycle stage. Additionally, based on the forward scatter measurements, we were able to corroborate that *naa10Δ* cells were 20% larger than WT cells (Fig. S7C), agreeing with what has been reported earlier (PMID: 33535049).

Fig. S7. Yeast *naa10Δ* cells have similar cell cycle distribution to WT but are 20 % larger in size. Yeast cells were diluted from over-night preculture and grown in synthetic medium until OD 0.8-0.9, fixed and stained with SYTOX Green and analyzed by flow cytometry. (A) Cell cycle distribution profiles of WT vs *naa10Δ* cells. Shown is overlay of three replicates from one of three independent experimental setups. (B) Percentage of cells in G1, S or G2

phase from three replicate experimental setups with seven replicate samples in total. Data are presented as mean \pm s.d. Error bars show standard deviation. Differences among WT vs *naa10Δ* cells was analyzed for all stages using two-tailed t-test with unequal variance (G1, $p=0.4$; S, $p=0.8$; G2, $p=0.2$). (C) Corresponding forward scatter histogram plot for data in A. Dashed lines indicate a right-shift for the *naa10Δ* peak vs WT. C'. Median FSC-A value from three replicate experimental setups with seven replicate samples in total. *** indicate $p=0.00017$ using two-tailed t-test with unequal variance.

2.-Since the authors extensively profile the N-terminome in both conditions and have a good idea of which are the substrates of NatA it would be worth to specifically represent the turnover and thermal stability changes of this group of proteins to better grasp the global changes.

We thank the reviewer for this suggestion. We have included a supplementary figure (Fig. S8) in which the identified NatA substrates are highlighted within the protein turnover vs thermal stability space. Reassuringly, gene ontology enrichment analysis of the two classes of NatA substrates mapped to functional interaction networks using STRING 11.5 (PMID: 33237311) showed similar results as our previous analysis described in Figure 4 and Figure 6. Line (407-411)

Fig. S8. Landscape of the NatA substrates within the thermostability and degradation space in NatA defective yeast cells. (A) Scatterplot representing the overlap between protein thermostability and degradation. Black lines delineate the minimal fold change ($s_0, 0.1$) in each dimension. Melting temperature and degradation difference distribution are shown in the top and right plot border respectively. Proteome and NatA substrates are colored in gray and green

respectively. **(B)** String networks of the significantly enriched GO terms (BP: Biological Process; MF: Molecular Function and CC: Cellular component) within the Non-stable (Blue) and the Non-stable/Fast turnover NatA substrates (Violet) are shown.

3.-Reference 69 should be the original TPP paper: PMID: 25278616

We have corrected it. From Savitski *et.al.*, 2018 to Savitski *et.al.*, 2014.

Reviewer #2 (Remarks to the Author):

The manuscript entitled ‘Loss of N-terminal acetyltransferase A activity induces thermally unstable ribosomal proteins and increases their turnover’ by Guzman and coworkers describes the impact of absent N-terminal acetylation (NTA) of nascent chains on global protein abundance, thermal stability, and ubiquitination rates in the unicellular model eukaryote yeast.

The study is well designed and considers that N-terminal acetyltransferase A (NatA) depleted yeast mutants grow slower than the wild type affecting the dilution of proteins by cell division. This is a unique selling point of this study and allows a better understanding of the physiological consequences of absent NTA on NatA substrates. The authors combined state-of-the-art global proteomics approaches addressing proteome abundance, NTA frequency, turnover rates (pSILAC), thermal stability (TPP-DIA combined with ITSA), and the ubiquitinome (DiGly peptide enrichment) to dissect the direct impact of lowered NatA substrate acetylation from secondary effects at the global scale. Such secondary effects cannot be avoided by the study design since the ribosome-tethered NatA addresses approximately 40% of the yeast proteome.

In contrast to previous studies in yeast, the authors could demonstrate a stabilizing impact of NTA on NatA substrates. These findings are adequately discussed within the context of previous work in yeast and other eukaryotes. Furthermore, the authors link the absent acetylation of NatA substrates to stronger ubiquitination and faster protein degradation. This aspect could even be strengthened in the manuscript (see suggestions/minor concerns). In the case of ribosomal NatA substrates, absent acetylation decreases the thermal stability of these proteins, which might be causative for the observed increased turnover. Remarkably, the enhanced turnover of NatA substrates was not accompanied by decreased protein abundance, which has also been shown for selected NatA substrates in other eukaryotes. However, the obtained findings are novel and far from being of confirmative nature since they uncover a so-far unknown detailed view of the protein fate at the global scale in response to NTA. The authors provided compelling evidence for their claims. This reviewer did not spot any flaws in the experimental planning or the interpretation of the large data sets.

Suggestions / minor concerns

1.-Title: The title should specify the organism used in the study since it is currently debated if this mode of action is conserved in other opisthokonts.

We have added the *Saccharomyces cerevisiae* to the title.

2.-Page 5: This analysis revealed that the free Nt-peptides have a significantly faster turnover rate compared to the Nt-acetylated peptides... -> A separate supplemental file listing the corresponding proteins and their predicted/known subcellular localization would be helpful for interested readers who are not proteomics experts.

We thank the reviewer for pointing out that a table containing fast turnover rate Nt acetylated peptides and their subcellular localization would be helpful for interested readers. We have now included such a list containing the requested data as a new Supplementary table 3.

3.-Page 2: , drought stress -> replace reference 7 with reference 32

We have corrected it. Reference 7 to 32. Linster *et.al.*, 2022 to Linster *et.al.*, 2015

4.-Page 10 ...As expected, the majority of the fast turnover proteins were more ubiquitinated in the *naa10delta* strain.... -> The authors should provide information on the identity of proteins and their subcellular localization (e.g., membrane association) in a separate supplemental file and shortly discuss what this means for degradation of the proteome via autophagy or the UPS. Potentially, recent findings by Shen and coworkers (Cell Reports, 2021) on autophagy induction in yeast by absent NatB might be discussed.

As mentioned in the answer to the previous question by the reviewer above, we have created an additional supplementary table in which this information is included. Following the advice of the reviewer, we now mention and discuss the identity and the role of selected ubiquitination events in UPS and autophagy in the results and discussion sections, respectively.

Results section, line (439:443), now reads:

Additionally, lysine residues that have been reported not to be accessible in the structure of the mature ribosome and common substrates of *TOM1*⁶⁰, as well important ubiquitination events related to RQC (ribosome-associated protein quality control) and NRD (non-functional rRNA decay) pathways such as RPS3-K212^{76,77} and RPL25-K74^{78,79} were found in our analysis.

Discussion section, line (496:503), now reads:

Moreover, ubiquitination events linked to RQC and NRD pathways (RPS3-K212^{76,77} and RPL25-K74^{78,79} respectively) were found in the *naa10Δ* ubiquitinome. Although, we observed autophagy markers such as ATG19 and ATG38 upregulated in *naa10Δ* strain, it has been reported that RPL25-K74 ubiquitination event prevents the degradation of the 60S by ribophagy. Therefore, it is very likely that degradation of ribosomal proteins in *naa10Δ* strain occurs mostly through the UPS system. However, the observed impairment of mitochondrial ribosomal protein degradation can be explained by the critical role of NatA in mitophagy⁸².

5.-Page 10 with recent studies in other species -> add Gong et al (Mol. Plant 2022)

We have added the reference Gong, et al., 2022

6.-Page 10, as well as protein inhibitors -> protein translation inhibitors

We thank the reviewer for noticing this and we have corrected it accordingly in manuscript text from “protein inhibitors” to “protein translation inhibitors”

Reviewer #3 (Remarks to the Author):

This manuscript by Guzman et al. studies the functional impacts of Nt-acetylation on the protein stability in yeast by the proteomics approaches. To investigate the effect of lacking Nt-acetylation on a proteome-wide scale, authors employ the *naa10Δ* strain, which leads to abolishing the NatA activity, for all proteomic analyses. The pulsed-SILAC analysis shows that the degradation of ribosomal proteins is increased in the *naa10Δ* strain and the thermal proteome profiling reveals the reduction of thermostability of ribosome proteins, suggesting that lack of Naa10 promotes the defective or delayed folding of ribosomal proteins, which increases the probability of their degradation.

The proteomic analyses are well performed and confirmed previous observations. However, the weak point of this paper is that the results presented in this paper provide just the resource of the proteome in *naa10Δ* strain, and there is no functional analysis of the N-acetylation of ribosomal proteins.

1.-The authors focus on the stability of the ribosomal proteins and claim that the lack of Naa10 promotes the defective or delayed folding of ribosomal proteins and consequently, increases the probability of their degradation. However, to address the functional role of N-acetylation of ribosome proteins, the authors should have addressed how N-acetylation is involved in the folding of the ribosomal proteins. For example, many ribosomal proteins have their unique chaperone, which helps the protein folding, the import into the nucleus, and the assembly into pre-mature ribosome particles and so on. Is the N-acetylation involved in such chaperone function?

The authors should have at least investigated and discussed the role of N-acetylation on the fate of ribosome proteins from co-translational folding to assembly step into pre-mature ribosome; northern blotting of premature rRNA is not enough to discuss the effects on the ribosome biogenesis. When and where the ribosomal proteins are degraded? co-translational ? before or after import into nucleolus ? or after assembled into ribosome ?

We thank the reviewer for the constructive feedback and suggestions. Although, the questions raised by the reviewer are indeed interesting, they are beyond the scope of our study, which was to investigate the fate of NatA substrates and associated proteins at a global scale in response to NatA knockout. Nevertheless, to address some of the questions made by the reviewer, we have now mapped all the ribosome biogenesis factors described so far (PMID: 36762427) onto the different proteome dimensions that we have analyzed in our study (Figure below).

Fig. 1. Multilayered landscape of ribosome biogenesis factors in NatA defective yeast cells. (A) Differential expression profiling of the WT and *naa10Δ* strains in a volcano plot. (B) Comparison of the *naa10Δ* and WT systems as a volcano plot to identify significant changes in protein turnover rates. (C) Comparison of the WT and *naa10Δ* strains in a volcano plot to identify changes in protein melting temperature. (D) Comparison of the WT and *naa10Δ* strains as a volcano plot to identify newly synthesized protein abundance significant changes. In all presented dimensions the significant regulated proteins at 1 % and 5% false discovery rate (FDR) are delimited by dashed and solid lines respectively (FDR controlled, two-sided *t*-test, randomizations = 250, $s_0 = 0.1$).

This multilayered landscape of ribosome biogenesis factors (RBFs) showed that multiple RBFs are NatA substrates and follow the same trends as the ribosomal proteins within the thermal stability and turnover rate space. Briefly, the differential expression analysis between WT and *naa10Δ* strains harvested at same OD (Fig. 1A), showed that RBF are down regulated in KO. In contrast, RBF have a faster turnover rate as well as lower

thermal stability (Fig. 1B-C). Reassuringly, the global change of newly synthesized proteins WT and *naa10Δ* strains (Fig. 1D) revealed that the fast degradation of RBF is compensated by higher synthesis in *naa10Δ* strain. This observation is aligned with the northern blot results showed in Fig. S3A-C and normal 60S/80S ratio observed in polysome profiling experiments in previous investigations (PMID: 33535049, PMID: 21184851). The fact that RBFs such as TSR4 and NOP12 are NatA substrates, which are thermally unstable and fast degraded, suggests that newly synthesized ribosomal proteins and ribosome precursors can be defectively folded (PMID: 21811236, PMID: 31062022). However as stated in line (297:300), the direct effect of Nt-acetylation on the ribosome is challenging to address since the fast degradation of ribosomal proteins in *naa10Δ* strain could be a consequence of a dysfunctionality caused by the lack of Nt-acetylation in RBF proteins or constituent ribosomal proteins.

Our datasets may also provide answers to when and where ribosomal protein degradation is taking place. The fact that Tom1, the E3 ubiquitin ligase in yeast that anchors a ubiquitin-proteasome system quality control pathway to eliminate ribosomal proteins that fail to assemble into ribosomes and are degraded in the nucleus/nucleolus, as well as other proteins related with the ubiquitin-proteasome system (UPS) such as UFD2 and RSP5 are upregulated in the KO strain strongly suggests that the degradation of ribosomal proteins is carried out by the UPS system. This hypothesis aligns well with the ubiquitination analysis showed in Fig. 7B, where lysine residues reported as concealed in the mature ribosome were found to be ubiquitinated. Moreover, ubiquitination sites important for the ribosome quality control pathway such as RPS3 K212 and RPL25 K74 were found in *naa10Δ* ubiquitinome (Table S7). Briefly, the RPL25 K74 ubiquitination site protects ribosomes from autophagy (PMID: 24616224, PMID: 32039200). Likewise, RPS3 K212 ubiquitination site is important for the RQC and NCD pathway and it is enhanced by starvation and translation inhibition (PMID: 29147007). Finally, since TOM1 is an ubiquitin ligase that restricts the accumulation of overexpressed ribosomal proteins through the ERISQ pathway (PMID: 27552055), we decided to create a double KO strain (*naa10Δ tom1Δ*) to try to rescue ribosomal proteins from proteasomal degradation (Fig. 2). Although only four ribosomal proteins from the large subunit were found to be upregulated in the *naa10Δ tom1Δ* compared to *naa10Δ*, this result suggests that the degradation of ribosomal proteins is carried out by multiple members of the ubiquitin-proteasome system, which aligns well with our previous observations.

These results were summarized in the supplementary figure showed below (Fig. S9) and described in the results section, lines (445-452), which now reads:

To test this hypothesis, we created a double KO strain (*naa10Δ tom1Δ*) and tried to rescue ribosomal proteins from proteasomal degradation (Fig.S9A-B). Although only four ribosomal proteins from the large ribosome subunit (RPL7, RPL4, RPL42 and RPL37) were upregulated in the *naa10Δ tom1Δ* compared to *naa10Δ* strain, this result suggests that additional ubiquitin ligases such as UFD2, UFD4 and UBR1 as well as proteasomal proteins, which were consistently up-regulated in the different dimensions of the *naa10Δ* proteome, are likely involved in the fast turnover of ribosomal proteins in NatA lacking cells.

Fig. S9. Effect of Tom1 on NatA defective yeast cells.

(A) Volcano plot displaying the \log_2 fold change against the t test–derived $-\log_{10}$ statistical P value for all proteins differentially expressed between WT and *naa10* Δ strains. (B) Same as A, but comparing the *naa10* Δ and *naa10* Δ *tom1* Δ strain proteomes. Significant regulated proteins at 1 % and 5% false discovery rate (FDR) are delimited by dashed and solid lines respectively (FDR controlled, two-sided t -test, randomizations = 250, $s_0 = 0.1$).

2.-The authors claim that the *naa10* Δ strain shows increased sensitivity to the protein translation elongation inhibitor cycloheximide and caffeine (Fig. 4D). However, the cell growth of both wild-type and *naa10* Δ are impaired by the addition of cycloheximide and caffeine with the same extent; there is no difference between wild-type and *naa10* Δ strain.

We agree with the reviewer that our original data in Figure 4D did not show a clear difference in growth between WT and *naa10* Δ strains on the agar plates with 0.1% caffeine and 0.1 μ g/mL cycloheximide. Thus, it does not correlate with previous data (Arnesen et al PNAS 2009, PMID 19420222). However, the yeast strains used in the 2009 paper have a slightly different genetic background as compared to the yeast strains employed in the current study and this may cause the difference in response. We also considered that there might have been a suboptimal drug application (dose etc) for the current setup. We have therefore optimized conditions further and based on this we found that for the current strains a doubling of the concentrations of caffeine and CHX could best display the phenotype. 30°C condition was repeated for "loading control". Consequently, we have updated Figure 4D with this new experiment:

Figure 4. D. Phenotypic growth of *S. cerevisiae naa10Δ* in the presence of various stressors. The indicated yeast strains were grown to early log phase and serial 1/10 dilutions containing the same number of cells were spotted on various media. 30 °C, incubated for 3 days on YPD at 30 °C; Caffeine, incubated for 3 days on YPD+0.2% caffeine; and CHX, incubated for 7 days on YPD+0.2 μg/mL cycloheximide. Results are representative of at least three independent experiments.

Reviewer #4 (Remarks to the Author):

In this manuscript Guzman et al. present a series of high-quality proteomics experiments to study the role of the main eukaryotic enzyme, N-terminal acetyltransferase (NatA), that mediates protein N-terminal acetylation in *Saccharomyces cerevisiae*. The sounded results allowed the authors to conclude that in yeast NatA-dependent N-terminal acetylation plays a role in protein stability and protein quality control in order to maintain cellular proteostasis, as it has been suggested already in plants and mammalian cells.

The authors started with a global and almost completed protein expression profiling comparing WT and Naa10Δ (NatA KO) strain proteomes and found significant differences between the two proteomes, specifically ribosomal proteins increased in the KO strain. Then, they performed a pulse SILAC experiment to compare half-lives of proteins between WT and Naa10Δ observing a general increased turnover of proteins, also ribosomal, in the NatA KO strain implicating thus Nt-acetylation in protein stabilization. With thermal proteome profiling and isothermal shift assays, the authors further connected the increased ribosomal protein turnover to decreased protein stability suggesting defects in folding or interactions. Finally, by using enrichment of ubiquitinated peptides combined with quantitative mass spectrometry the authors found lack of NatA in yeast leads to an increased ubiquitination of most proteins, and particular ribosomal proteins, which would lead to an increased degradation via the proteasome, explaining thus the divergence of the proteomes between wt and NatA KO strains.

In summary, this study revealed that in yeast NatA-mediated Nt acetylation in yeast is critical for protein stabilization, including ribosomal proteins, most likely by promoting optimal folding and/or supporting protein-protein interactions by avidity.

These findings are of great general interest and represent an important contribution to the understanding of how organism control cellular proteostasis beyond state of the art which merits publication in Nature Communications. Conclusions are drawn by sounded experimental data and robust statistical analysis. Overall, this is a well written manuscript with data and methodology aligning with recent advances in the field. I therefore recommend publication of this manuscript with very minor revisions, see points below.

Minor points:

1.-Fig. 1SB: Under the third and fourth column, there is written Naa20 and should be Naa15. There is no indication about which statistical test and p-value has been performed in order to get the significant differences between WT and Naa10Δ. Significant differences should be indicated as well in the figure.

We have corrected it and added the requested information

Fig. S1. Lack of NatA activity (*naa10Δ* cells) elicit phenotypic changes by down-regulation of ribosomal proteins and up-regulation of autophagy markers. (A) Upper: WT and *naa10Δ* strain growth curves monitored by Optical density at a wavelength of 600 nm (OD). Cell cycle time is indicated for each condition (Tcc). Bottom: Phenotypic growth of *S. cerevisiae naa10Δ* at different temperatures. The indicated yeast strains were grown to early log phase and serial 1/10 dilutions containing the same number of cells were spotted on various media and imaged the six following days. 30 °C, incubated for 3 days on YPD at 30 °C; 15 °C, incubated for 6 days on YPD at 15 °C; 37 °C, incubated for 6 days on YPD at 37 °C. Results are representative of three independent experiments. (B) Relative abundance of the detected N-terminal acetyltransferase (NAT) subunits. Each identified protein was observed in at least two of the three biological replicates. Error bars represent standard deviation. Mean ± standard deviations are shown. Statistical significance was assessed using two-sided t-test, multiple test correction according to Benjamini-Hochberg, ns = P > 0.05, *P ≤ 0.05, **P ≤ 0.01, ***P ≤ 0.001, ****P ≤ 0.0001 (C) Differential expression profiling of the WT and *naa10Δ* strains in a volcano plot. Significant regulated proteins at 1 % and 5% false discovery rate (FDR) are delimited by dashed and solid lines respectively (FDR controlled, two-sided t-test, randomizations = 250, s0 = 0.1) (D) GSEA-based KEGG pathway enriched analysis. Significance threshold set at FDR > 0.05.

2.-Fig. 1 The abbreviation RP, RPL and RPS are quite intuitive but should be also spelt, for example in the legend of Figure 1.

We have corrected it. We added the sentence: “Ribosomal proteins from the large (RPL) and small subunit (RPS) are colored in blue and green respectively”

3.-Page 4 last line first paragraph is referenced to Fig. S1C, but should be Fig. S1B

We have corrected it. Fig. S1C to Fig. S1B

4.-Page 5, second paragraph a bracket around Fig. 2C is missing.

We have corrected it. From WT strain Fig. 2C) to WT strain (Fig. 2C)

5.-Figure 2: please mentioned in the figure legend what Tdil is.

We have corrected it. The WT and *naa10A* calculated dilution constant (Kdil) and dilution time (Tdil; cell double time) are shown in red and blue, respectively.

6.0-Figure 3:

(A) In the manuscript text the data of this panel is clear but the graph in the figure is not easy to interpret. Please, revise graph and figure legend and make it clear.

Also, the legend indicates twice that the Tcc is marked by an arrow, but this is not clearly visible. Please, revise this as well.

We thank the reviewer for the input on how to improve the Figure 3, which contains the evaluation of the cell cycle impact on protein half-life estimation. To make the description clearer, we added a supplemental figure (Fig. S3) describing the strategy implemented line (22).

Fig. S2 Dilution constant effect on protein half-lives. A) Upper: A comparison between WT half-lives (x axis) and calculated half-lives when the $K_1 = 1$ ($K_{dil_{WT}} = K_{dil_{KO}}$, y axis). Two regions are delimited (dashed line) as function of the cell cycle time (Black dot). Bottom: Diagram showing the time-domain RIA incorporation into *naa10Δ* and WT system proteins when the $K_{dil_{WT}} = K_{dil_{KO}}$, $K_1 = 1$. B) and C) Same as A, but when $K_{dil_{WT}} > K_{dil_{KO}}$, $K_1 < 1$ and $K_{dil_{WT}} < K_{dil_{KO}}$, $K_1 > 1$ respectively.

In addition to the new supplementary figure S3, we also improved Fig. 3's legend and graph. We have indicated the T_{cc} as a black dot.

6.1.-

(B) This figure has been done for the whole proteome. How does the curves look focusing only on NatA targets? Is the shift of the curve between WT and *Naa10Δ* more pronounced? This would be an interesting point to add to the manuscript.

We add the requested comparison in Fig. S3A & B (presented below). Additionally, we referenced this observation in the new version of the manuscript (line 218-22)

Fig. S3. NatA substrates in the *naa10Δ* strain proteome shows faster normalized degradation rates compared to the full yeast proteome.

(A) Cumulative frequency plot of the normalized turnover rate (Kdeg/Kdil) determined in *naa10Δ* and WT system of NatA substrates (WT, blue; *naa10Δ*, red). Kolmogorov-smirnov test, KS $P = -3.64 e-05$. (B) Cumulative frequency plot of the normalized turnover rate (Kdeg/Kdil) of NatA substrates and full proteome determined in *naa10Δ* system (*naa10Δ* proteome, blue; *naa10Δ* NatA substrates, red). Kolmogorov-smirnov test, KS $P = -3.64 e-05$. (C) Violin plot of normalized turnover rates of WT and *naa10Δ* NatA substrates compared to their corresponding N-terminome. Statistical significance was assessed using two-sided Wilcoxon test, multiple test correction according to Benjamini-Hochberg, ns = $P > 0.05$, $*P \leq 0.05$, $**P \leq 0.01$, $***P \leq 0.001$, $****P \leq 0.0001$; box corresponds to quartiles of the distribution. Overlap between the N-terminome and proteome detected in the pSILAC, as well as the N-terminome acetylation status and NAT substrate class are shown on top. (D) Same as C, but comparing NatA substrates with full proteome in *naa10Δ* condition. Only NatA substrates having a serine (S) in second amino acid of annotated protein sequences retrieved from the UniProt database were considered. (E-F) Same as D but NatB (ME, MD, MQ and MN) and NatC/E (ML, MK, MF and MY) substrates were compared against the full proteome in *naa10Δ* condition respectively.

6.2.-

(C+D) The comparisons of the whole proteome set to the whole N-terminome dataset does not seem meaningful. Here, the individual protein overlap between the proteome and Nt-peptides is not stated. Would one not expect that if the protein is degraded then corresponding Nt-peptide is also degraded? The data of protein and N-termini should go hand in hand? Thus, it is not clear why to expect significant differences between proteome and Nt-terminome since the Nt-terminome should represent a quite random subset of the proteome. It is as well not indicated how many peptides are Nt-acetylated or free. I would suggest to divide the N-terminome in potential NatA targets (eventually including Naa50 targets as they might be affected as well) and targets of NatB and C and then compare the proteome data. Alternative, divide the proteome as well in subgroups based on NatA targets and/or other Nats.

We thank the reviewer for input regarding Fig 3C & D. We updated the figure 3 accordingly (presented below). Briefly, we clarified in the manuscript text that the Nt-acetylation turnover rates were directly inferred from the pSILAC experiment (Not enriched). Therefore, it is expected that NatA substrates were dominant due to high abundance and frequency in the proteome. (Line 218-222). Additionally, we updated the Fig. 3 showing the Nt-peptides overlap with the corresponding proteome and the proportion of acetylated and non-acetylated peptides.

We also included in the Fig. S3 the comparisons of the NatA-C substrates against the proteome as the reviewer requested. These findings also corroborate the results from Figure 4, which now is explained in lines (275-277).

Fig. 3. Lack of N-terminal acetylation due to deletion of *NAA10* promotes protein degradation of NatA substrates in the yeast proteome. (A) A comparison between determined half-lives in WT system ($T_{1/2}$) and modeled effect of WT half-lives ($T^*_{1/2}$) with a reduced dilution constant ($K_{dil_{KO}} = -0.26$). Red line represents the predicted half-lives due to the lack of *naa10* and blue line the determined half-lives in WT. Cell cycle time ($T_{cc} = 1.9$ h) of the WT system is marked by an arrow (Black dot). Data were modeled using the equation show on top and calculated according to Eden, et.al⁸². (B) Cumulative frequency plot of the normalized turnover rate (K_{deg}/K_{dil}) determined in $naa10\Delta$ and WT system (WT, blue; $naa10\Delta$, red). Kolmogorov-smirnov test, KS $P = -2.2 e-16$. (C-D) Violin plot of normalized turnover rates of WT and $naa10\Delta$ proteome compared to their corresponding N-terminome. Statistical significance was assessed using two-sided Wilcoxon test, multiple test correction according to Benjamini-Hochberg, ns = $P > 0.05$, * $P \leq 0.05$, ** $P \leq 0.01$, *** $P \leq 0.001$, **** $P \leq 0.0001$; box corresponds to quartiles of the distribution. Overlap between the N-

terminome and proteome detected in the pSILAC, as well as the N-terminome acetylation status and NAT substrate class are shown on top. (E) Same as C-D, but comparing the N-terminal acetylated peptides of the NatA type detected in the WT and their corresponding unmodified N-terminal peptides detected in *naa10Δ* cells.

7.-Ref 51 is not upper case.

We have corrected it. (51 → ⁵¹)

8.-Fig 6D: it is not clear what the grey box with NA means.

We have corrected and added the following explanation to NA: no enriched terms at specified cutoff

9.-Page 9, paragraph 3 there is a typo “catalityncore”.

Thanks. We have corrected it. (catalityncore → catalytic core)

REVIEWERS' COMMENTS

Reviewer #1 (Remarks to the Author):

I am happy with the revision and recommend publication

Reviewer #3 (Remarks to the Author):

The revised manuscript has been improved significantly. However, if the authors address how N-acetylation is involved in the folding of the ribosomal proteins to regulate thermostability, it would provide more strong interest to a wider audience.

In addition, the ubiquitination of ribosome proteins is not linked to the correct events in the revised manuscript. The authors described as follows at line 496 ~; "ubiquitination events linked to RQC and NRD pathway (RPS3-K212 and RPL25-K74, respectively) were found in", however, RPS3-K212 and RPL25-K74 are linked to NRD and ribophagy, respectively. RPS20-K6 and K8 are linked to the RQC pathway. These must be correct before publication.

Reviewer #4 (Remarks to the Author):

The authors successfully addressed all the comments indicated in the review, I therefore consider this manuscript and the scientific findings to be accepted for publication in Nature Communications.

REVIEWERS' COMMENTS

Reviewer #1 (Remarks to the Author):

I am happy with the revision and recommend publication

Reviewer #3 (Remarks to the Author):

The revised manuscript has been improved significantly. However, if the authors address how N-acetylation is involved in the folding of the ribosomal proteins to regulate thermostability, it would provide more strong interest to a wider audience.

In addition, the ubiquitination of ribosome proteins is not linked to the correct events in the revised manuscript. The authors described as follows at line 496 ~; "ubiquitination events linked to RQC and NRD pathway (RPS3-K212 and RPL25-K74, respectively) were found in", however, RPS3-K212 and RPL25-K74 are linked to NRD and ribophagy, respectively. RPS20-K6 and K8 are linked to the RQC pathway. These must be correct before publication.

We express our gratitude to the reviewer for bringing this issue to our attention at line 496. The sentence has been corrected to:

We found ubiquitinated events associated to NRD and RQC pathways (RPS3-K212⁷⁸ and RPS20-K8⁸⁰ respectively) in the *naa10Δ* ubiquitinome. In contrast, despite observing upregulated autophagy markers such as ATG19 and ATG38 in *naa10Δ* strain, ubiquitination of RPL25-K74^{84,85} was observed. This ubiquitination event has been previously been reported to prevent the degradation of the 60S subunit by ribophagy.

Reviewer #4 (Remarks to the Author):

The authors successfully addressed all the comments indicated in the review, I therefore consider this manuscript and the scientific findings to be accepted for publication in Nature Communications.